# B cell expression of an enzymatic intermediary in ether lipid biosynthesis promotes antibody responses and germinal center size

Sung Hoon Cho[1,2,3]*[†], Marissa A Jones[4,5], Kaylor Meyer[1], David M Anderson[4,5], Sergiy Chetyrkin[4,5], M Wade Calcutt[4,5], Richard M Caprioli[4,5], Clay F Semenkovich[6], Mark R Boothby[1,2,3,7]*

[1]Department of Pathology-Microbiology-Immunology, Vanderbilt University Medical Center, Nashville, United States; [2]Vanderbilt Centerfor Immunobiology, Vanderbilt University Medical Center, Nashville, United States; [3]Vanderbilt Institute for Infection, Inflammation, and Immunology, Vanderbilt University Medical Center, Nashville, United States; [4]Vanderbilt University Medical Center, Nashville, United States; [5]Mass Spectrometry Research Center and Department of Biochemistry, Vanderbilt University Medical School, Nashville, United States; [6]Division of Endocrinology Metabolism and Lipid Research, Department of Medicine, Washington University, St. Louis, United States; [7]Cancer Biology Program, Vanderbilt University, Nashville, United States

*For correspondence:
sung.hoon.cho@emory.edu (SHC);
mark.boothby@vumc.org (MRB)

Present address: [†]Department of Medicine (Rheumatology), Emory University, Atlanta, United States

Competing interest: The authors declare that no competing interests exist.

## eLife Assessment

This study provides **useful** insights into the ways in which germinal center B cell metabolism, particularly lipid metabolism, affects cellular responses. The authors use sophisticated mouse models to **convincingly** demonstrate that ether lipids are relevant for B cell homeostasis and efficient humoral responses. The authors then conducted in vivo as well as in vitro experiments, thereby strengthening their conclusions.

**Abstract** The qualities of antibody (Ab) responses provided by B lymphocytes and their plasma cell (PC) descendants are crucial facets of responses to vaccines and microbes. Metabolic processes and products regulate aspects of B cell proliferation and differentiation into germinal center (GC) and PC states along with Ab diversification. However, there is little information about lymphoid-cell-intrinsic functions of enzymes that mediate ether lipid biosynthesis. Imaging mass spectrometry (IMS) results had indicated that concentrations of a number of these phospholipids were substantially enhanced in GC compared to the background average in spleens, but it was unclear if biosynthesis in B cells was a basis for this finding, or whether cell-intrinsic biosynthesis contributes to B cell physiology or Ab responses. Ether lipid biosynthesis can involve the enzyme PexRAP, encoded by the *Dhrs7b* gene. Using IMS and immunization experiments in mouse models with inducible *Dhrs7b* loss of function, we now show that B-lineage-intrinsic expression of PexRAP promotes the magnitude and affinity maturation of a serological response. Moreover, the data revealed a *Dhrs7b*-dependent increase in ether phospholipids in primary follicles with a more prominent increase in GC. Mechanistically, PexRAP impacted B cell proliferation via enhanced survival associated with controlling levels of ROS and membrane peroxidation. These findings reveal a vital role of this peroxisomal enzyme in B cell homeostasis and the physiology of humoral immunity.

## Introduction

The qualities and concentrations of antigen (Ag)-specific antibodies (Ab) are key elements of immunity and the pathophysiology of diverse conditions involving inflammation (*Geyer et al., 2021*; *McGettigan and Debes, 2021*; *Lu et al., 2018*). Abs are secreted by cells in the B lymphocyte lineage, but the exact sources are diverse (*Allman and Pillai, 2008*; *Martin and Kearney, 2000*; *MacLennan et al., 2003*; *Hsu et al., 2006*). Many derive from a subset termed B1 B cells, which are thought to have characteristics of innate responses (*Allman and Pillai, 2008*; *Martin and Kearney, 2000*), but progeny of follicular and marginal zone B cells (FoB and MZB, respectively) can differentiate into Ab-secreting plasma cells (PC) after activation and proliferation (*MacLennan et al., 2003*; *Hsu et al., 2006*; *Rajewsky, 1996*). Ab concentrations and affinity for Ag, as well as Ab isotype after class switching in a B cell, determine the efficacy of pathogen clearance (*Boothby et al., 2019*; *Rubtsova et al., 2013*; *Alemán and Rosales, 2023*; *Xu et al., 2012*). Importantly, the affinities of serum Ab for target Ag can increase over time after an immune exposure, especially after recurrent encounter(s) (*Rajewsky, 1996*; *Boothby et al., 2019*; *Elsner and Shlomchik, 2020*; *Good-Jacobson and Shlomchik, 2010*).

Many sources and forms of Ab can be protective, although in several autoimmune diseases these molecules and the somatic mutations that diversify an initial BCR repertoire drive pathology (*Shlomchik et al., 1990*; *Thomas and Hulbert, 1996*). The micro-anatomic structure termed the germinal center (GC) provides one mechanism to increase the spectrum of Ab affinities over time and repeated Ag exposure (*De Silva and Klein, 2015*; *Cyster and Allen, 2019*; *Victora and Nussenzweig, 2022*). Although T cell help-dependent diversification and PC production can occur independent from GC, results with different forms of vaccination support the practical importance of GC (*Guttormsen et al., 1999*; *Kim et al., 2022*). Current evidence holds that most Ig class-switching is executed before GC entry of proliferating B cells (*Roco et al., 2019*). GC reactions are initiated after B cells in a primary lymphoid follicle encounter Ag, migrate to interact with activated helper T cells after extensive proliferation (*Gitlin et al., 2014*). GC formation, size, and function depend on multiple factors. These include the efficiency of population growth for B cells in a phase before they enter and take on characteristics of GC B cells, and on homeostatic and differentiative processes while in the secondary follicle (reviewed in *Vinuesa et al., 2016*). Once in the GC, B cells cycle between light and dark zones, undergoing iterative cycles of selection. Proliferation and selection end in development as Ab-secreting plasma cells or memory B cells (reviewed in *Good-Jacobson and Shlomchik, 2010*; *Cyster and Allen, 2019*; *Victora and Nussenzweig, 2022*). In line with a selection process, substantial rates of B cell death have been measured in GC (*Mayer et al., 2017*). As such, the qualities and quantities of Ab elicited by immunization (or vaccination) depend on B cell proliferation and survival both before and during GC reactions.

Accumulating evidence indicates that metabolic reprogramming and intermediary metabolism play pivotal roles in survival, proliferation, differentiation, and function in the pre-immune populations (*Chapman and Chi, 2022*; *Bacigalupa et al., 2024*), including B cells (*Jellusova, 2020*; *Boothby et al., 2022*). Among core metabolic processes, several converge on oxidative metabolism, fed by glycolytic generation of pyruvate, anaplerotic use of amino acids such as glutamine, and fatty acid oxidation (reviewed in *Boothby et al., 2022*; *O'Neill et al., 2016*) Compared to resting populations such as naive B cells, rates of such metabolism in B cells appear to be increased after activation, including in GC B cells (*Cho et al., 2016*; *Weisel et al., 2020*; *Chen et al., 2021*; *Urbanczyk et al., 2022*; *Yazicioglu et al., 2023*; *Brookens et al., 2024* reviewed in *Boothby et al., 2022*). Such increases support the generation of substrates for anabolic processes in the dividing cell population and perhaps the energy demands thereof.

Mitochondria have been the focus of investigations exploring molecular mechanisms for oxidative metabolism to generate energy and substrates in lymphocytes during immune responses [reviewed in *Olenchock et al., 2017*; *Kong and Chandel, 2018*]. Oxidative phosphorylation in and dynamics of this organelle appear to be important in determining rates of somatic hypermutation that characterize GC B cell biology (*Chen et al., 2021*; *Urbanczyk et al., 2022*; *Yazicioglu et al., 2023*), albeit by a molecular mechanism that has not yet been identified. Among specialized functions of this highly dynamic subcellular organelle, mitochondria use the Krebs TCA cycle to feed an electron transport chain (ETC) and efficiently generate ATP. Along with other sources, mitochondria generate reactive oxygen species (ROS) via the ETC at a rate determined by structural and biochemical features

(*Addabbo et al., 2009*; *Zorov et al., 2014*). Rates of ROS production, release, and resolution require regulation at multiple levels because these labile species and their reactions with cellular molecules are important for both signaling and cell proliferation but undermine cell function or survival when excessive.

B cells in most GCs exhibit inadequacy of oxygen delivery relative to demands (*Cho et al., 2016*; *Jellusova et al., 2017*; *Li et al., 2021*; *Nakagawa et al., 2023*). Of note, genome-wide screening for genes that help cells adapt to such hypoxia revealed that peroxisomes and an ER enzyme involved in synthesizing plasmalogen ether lipids (EL) decrease the death of cells due to insufficient oxygen (*Jain et al., 2020*). Using an unbiased lipidomic analysis that combined imaging mass spectrometry (IMS) and techniques for identifying GC in sections of spleen from immunized mice, we found that local concentrations of a subset of EL were heightened in GC relative to other portions of the spleen (*Jones et al., 2020*). Ether phospholipids (EPL) have a fatty acid linked to the glycerol backbone with an alkyl or a vinyl ether bond at the sn-1 position (*Lodhi et al., 2015b*; *Lodhi et al., 2012*). Synthesis of ether phospholipids depends on the generation of precursors in peroxisomes and final processing in the ER (*Dean and Lodhi, 2018*; *Honsho and Fujiki, 2023*). However, EL biosynthesis is prominent in the liver, and in theory, these species or their precursors might distribute to tissues from a primary site of biosynthesis or from the diet (*Honsho and Fujiki, 2023*; *Werner et al., 2023*; *Dorninger et al., 2022*; *Kuerschner et al., 2012*; *Lodhi et al., 2015a*). Thus, the finding that secondary follicles had greater densities of some - but not all - ether phospholipids left several key questions unanswered (*Geyer et al., 2021*): Does the capacity to synthesize and increase ether lipids in B cell follicles - primary or secondary (i.e. GC) - affect antibody responses? (*McGettigan and Debes, 2021*) Are any of the ELs dependent on B-lineage-intrinsic metabolism to generate precursors or final products? Moreover, there is a debate whether EL or their plasmalogen subset promotes cell death or resistance, and whether inactivation of biosynthesis causes defects in establishment or maintenance of hematopoietic cells (*Lodhi et al., 2015b*; *Lodhi et al., 2015a*; *Dorninger et al., 2015*; *Cui et al., 2021*).

Peroxisomal Reductase Activating PPARγ (PexRAP, encoded by the *Dhrs7b* gene) catalyzes the reduction of alkyl-dihydroxyacetonephosphate (DHAP) to 1-alkyl-G3P (*Lodhi et al., 2015b*; *Dean and Lodhi, 2018*; *Honsho et al., 2020*), which then is transferred into endoplasmic reticulum (ER) and further metabolized to generate EL. Recent work notes that this enzyme may substantially function in the ER in addition to the peroxisome (*Honsho et al., 2020*). Widespread inactivation of *Dhrs7b* in mature mice decreased hepatic production of ether lipid species and reduced the population of erythrocytes and neutrophils (*Lodhi et al., 2015b*). This body of work indicated that PexRAP can be essential in some aspects of ether lipid metabolism and may function in hematopoietic cells, but was questioned based on inborn errors of metabolism affecting other aspects of ether lipid biosynthesis (*Dorninger et al., 2015*). Diverse functions of ether lipids in general, and the plasmalogen subset in particular, have been proposed or supported by prior work (reviewed in *Braverman and Moser, 2012*; *Gorgas et al., 2006*; *Jiménez-Rojo and Riezman, 2019*; *Rangholia et al., 2021*; *Bozelli et al., 2021*). The ether linkage may provide a sink for oxygen radicals, thereby acting directly to resolve the reactivity and contribute to ROS homeostasis, although there is debate as to the physiological impact of this chemical property (*Wallner and Schmitz, 2011*). Whether or not this lipid subset influences adaptive immunity or conventional lymphocyte lineages, however, is unclear.

Herein, we tested the hypotheses that (i) the ether lipid profiles in activated B cells depend on their expression of the biosynthetic enzyme PexRAP (alternatively referred to as acyl/alkyl-DHAP reductase [ADHAPR]; *Honsho et al., 2020*), and (ii) this enzyme promotes B cell physiology. To do so, we combined conventional lipidomics, Imaging Mass Spectrometry (IMS), and immunization experiments with genetic models in which loss of function for *Dhrs7b* is induced in adult mice. We provide evidence that PexRAP impacts B cell proliferation via enhanced survival associated with controlling levels of ROS and membrane peroxidation. The results also indicate that beyond an impact on B cell homeostasis, GCs are reduced in its absence, and the magnitude and affinity maturation of a serological response depend on B cell expression of PexRAP. Taken together, these findings support a function of B-cell-intrinsic biosynthesis of ether lipids in B lymphocyte homeostasis, the production of Abs, and in promoting GC reactions.

# Results

## PexRAP promotes homeostatic maintenance and proliferation of B cells

By use of IMS, we had reported that at least a dozen ether phospholipids - including plasmalogens - were enriched in splenic germinal centers of immunized mice when compared to the rest of the tissue (*Jones et al., 2020*). Previous work provided evidence that these lipid molecules were important for the homeostasis of short-lived neutrophils (*Lodhi et al., 2015b*). However, this effect was questioned based on hematological data from patients with inborn errors of metabolism that affect ether lipid biosynthesis (*Dorninger et al., 2015*). Moreover, the impact of these molecules on adaptive immune cells or functions is not known. Accordingly, we sought to test for effects of an intervention that interferes with a key biosynthetic pathway used for endogenous synthesis of these molecules. To do so, we started with analyses using induced loss of function for *Dhrs7b*[f/f] and the widely expressed *Rosa26-CreER*[T2] transgene (*Friedrich and Soriano, 1991*; *Shapiro-Shelef et al., 2005*) used in *Lodhi et al., 2015b*, which reduced hepatic generation of a subset of ether lipids and acts in all hematopoietic cells as well.

We modeled initial experiments on the published work with imaging mass spectrometry (*Jones et al., 2020*). Mice were treated with tamoxifen using conditions less intense than a regimen shown to have no effect on pre-immune splenic and B lineage populations (*Higashi et al., 2009*; *Uhmann et al., 2009*), immunized with SRBC [akin to the analyses in *Cho et al., 2016*; *Brookens et al., 2024*; *Jellusova et al., 2017*; *Jones et al., 2020*], and analyzed 7 days after immunization (*Figure 1A*). Functional inactivation of *Dhrs7b* in T and B lymphocytes was confirmed by western blot analysis using anti-PexRAP Ab (*Figure 1B*). The frequencies and numbers of CD19[+] B220[+] B cells in *Dhrs7b*[f/f]; *Rosa26-CreER*[T2] - only a small minority of which would have been activated by the immunization - were approximately 0.6 the values for controls (*Dhrs7b*[+/+]*Rosa26-CreER*[T2], i.e. wild-type locus; *Figure 1C and D*). In contrast to B cell populations, TCRβ[+] CD4[+] T cells and TCRβ[+] CD4[-] CD8 T cells were intact in tamoxifen-treated and SRBC-immunized *Dhrs7b*[f/f]; *Rosa26-CreER*[T2] mice (*Figure 1—figure supplement 1*, panels A-C). These results suggest that *Dhrs7b* is required for fully maintaining the B cell population but indicate that the enzyme is dispensable for the main sets of conventional T cells. In light of the earlier finding that a subset of ether lipids was more prevalent in secondary follicles, we analyzed GC in these samples. Microscopy and flow cytometry analyses after immunofluorescent staining found that less GC B cells were present after immunization of *Dhrs7b*[f/f]; *Rosa26-CreER*[T2] mice, that is, less IgD[-] CD95[+] GL7[+] (GC-phenotype) B cells (*Figure 1E–I*; *Figure 1—figure supplement 1*, panel D). Quantitation of the immune fluorescence micrography with spleens from immunized mice revealed that the numbers of GC per spleen were halved in *Dhrs7b*Δ/Δ mice compared with controls (*Figure 1I*; *Figure 1—figure supplement 1*, panel E). The sizes of those GCs that did form were substantially reduced as well (*Figure 1I*; *Figure 1—figure supplement 1*, panel F). The lower number of B cells found in immunized mice - while a less profound effect than the reduction in GC - tempers the result, and it was possible that the function of helper CD4 T cells was impaired even though their numbers were not affected. Consistent with this hypothesis, the frequencies of Tfh and GC-Tfh cells were lower in *Dhrs7b*Δ/Δ mice compared with controls (*Figure 1—figure supplement 1*, panels G-I). While not addressing whether or not B cells are part of the requirement for PexRAP in promoting GC, these results indicate that *Dhrs7b* is necessary for a normal-sized germinal center response.

To determine if there are B-lineage-specific functions of PexRAP, we used a conditionally active Cre transgene that is expressed specifically in mature B cells (*Khalil et al., 2012*). *Dhrs7b*[f/f], *huCD20-CreER*[T2] mice and *huCD20-CreER*[T2] were treated with tamoxifen to induce Cre-mediated recombination of the floxed alleles after initial establishment of normal B cell populations (*Figure 2A*). Western blot analyses showed an almost complete absence of PexRAP protein from B cells purified from tamoxifen-treated *Dhrs7b*[f/f]; *huCD20-CreER*[T2] (hereinafter, B-cell-specific *Dhrs7b*Δ/Δ or *Dhrs7b* D B, shown as *Dhrs7b*Δ/Δ−B) mice compared to similarly treated *huCD20*-CreER[T2] controls (*Figure 2B*). The enzyme encoded by this gene catalyzes the synthesis of a lipid precursor to many - but not all - plasmalogens and other ether phospholipids (*Lodhi et al., 2015b*; *Lodhi et al., 2012*; *Dean and Lodhi, 2018*), and *Dhrs7b* deficiency impacted neutrophil survival (*Lodhi et al., 2015b*). When we tested if *Dhrs7b* inactivation affected the steady-state B cell population, the frequencies and total numbers of CD19[+] B220[+] B cells in viable lymphocyte gates were about 20% lower in tamoxifen-treated B-cell-specific *Dhrs7b*Δ/Δ mice compared to tamoxifen-injected, *huCD20*-CreER[T2], *Dhrs7b*+/+controls (*Figure 2C*). With loss of function after production of mature B cells, that is inactivation of *Dhrs7b* at ages over 6 wk,

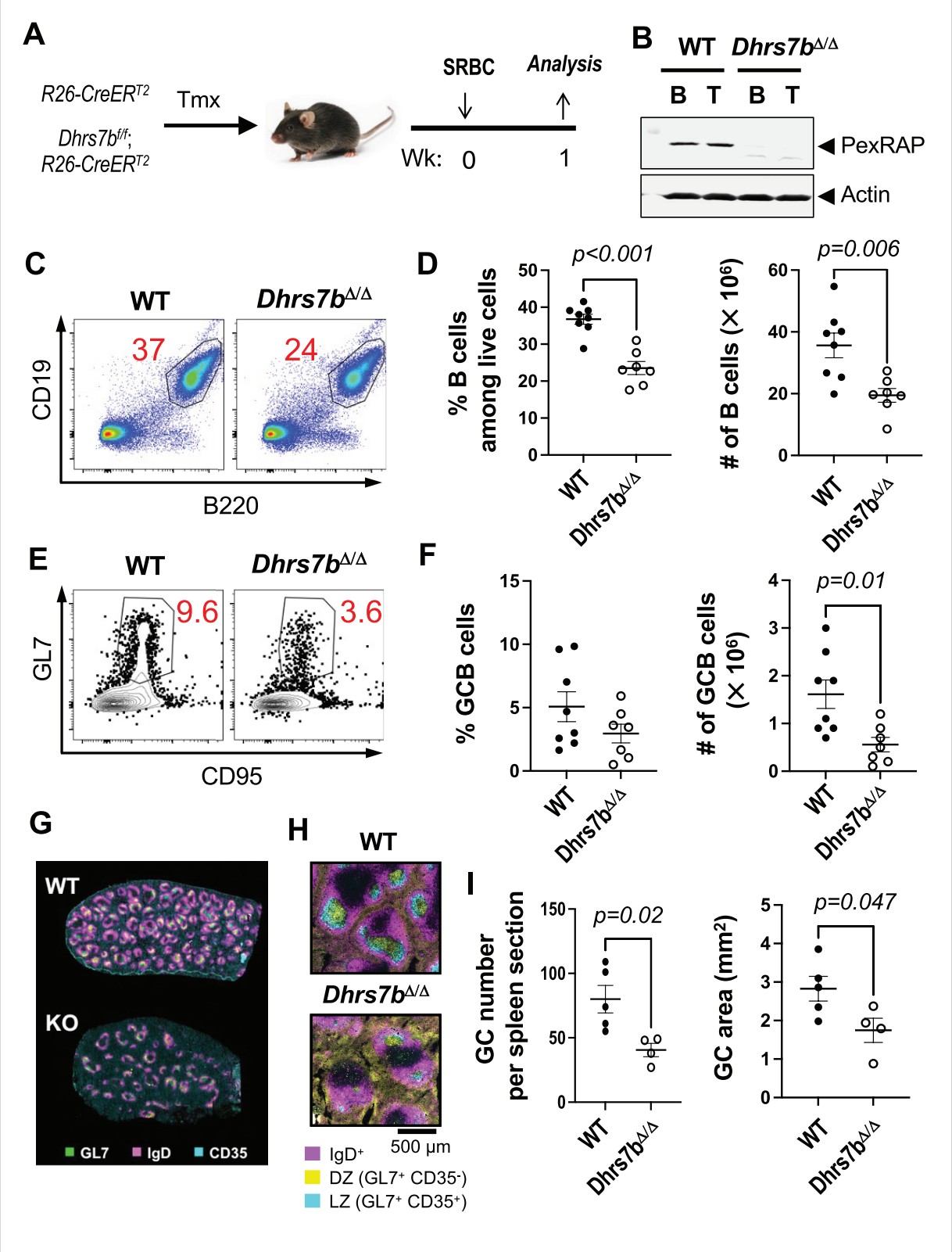

**Figure 1.** PexRAP promotes GC response. (**A**) Schematic of immunization with SRBC after inactivation of *Dhrs7b* in adult mice. Mice (*Rosa26*-CreER^T2^; *Dhrs7b^+/+^, or Rosa26*-CreER^T2^; *Dhrs7b^f^*) were treated with tamoxifen, immunized with SRBC, and harvested 1 week after immunization, as described in *Materials and methods*. (**B**) Deletion efficiency of *Dhrs7b* conditional alleles. B and T lymphocytes were isolated from spleens of *Rosa26*-CreER^T2^ or *Dhrs7b^f/f^;Rosa26*-CreER^T2^ mice after in vivo tamoxifen injections followed by immunization with SRBC. WT and *Dhrs7b^Δ/Δ^* were analyzed

*Figure 1 continued on next page*

*Figure 1 continued*

by immunoblotting with antibodies directed against PexRAP (protein product of *Dhrs7b*) and actin (internal loading control). (**C, D**) PexRAP acutely regulates B cell numbers. Representative flow plots of viable splenic B cells (**C**) and aggregate data for three biologically independent replicate experiments (**D**) (n=8 WT and 7 cKO). For data on T cells, see *Figure 1—figure supplement 1A–C*. (**E, F**) Effect of PexRAP on GC B cell response. Flow plots of GL7+ CD95+ GC B cells in the gate for viable IgD-negative (IgD^neg), dump-negative B cells (**E**), and aggregated frequencies and numbers of GC B cells, as indicated, in the three replicate experiments (**F**). (**G, H**) PexRAP impact on GC response. After tamoxifen injections, *Dhrs7b*-deficient mice [*Rosa26*-CreER^T2; *Dhrs7b^f/f* (*Lodhi et al., 2015b*)] and WT controls (*Rosa26*-CreER^T2) were immunized with SRBC and analyzed 7 days later, as in A–F. Data on gating strategy, GC counts per unit area and LZ area are in *Figure 1—figure supplement 1D–F* and on Tfh cells in *Figure 1—figure supplement 1G*. (**G, H**) Shown are representative images from immunofluorescence staining of spleens with the indicated Ab in two independent experiments (5 WT vs 4 *Dhrs7b* cKO, i.e. *Dhrs7bΔ/Δ*), showing a low-power overview with many follicles (**G**) and a representative higher-magnification image to better delineate primary and secondary follicles (**H**). (**I**) Quantitation of number (left panel) and size (right panel) of GC in spleen sections, with GL7+, IgD^neg areas that include both CD35+ (LZ) and CD35^neg (DZ) areas. The Mann-Whitney U test was used to calculate p values. Additional information is in *Figure 1—figure supplement 1*.

The online version of this article includes the following source data and figure supplement(s) for figure 1:

**Source data 1.** Raw images for the immunoblots shown in *Figure 1B*.

**Source data 2.** Uncropped images for the immunoblots shown in *Figure 1B*, with relevant bands labelled.

**Figure supplement 1.** PexRAP is dispensable for T cell numbers, but impacts the Tfh cell population prevalence.

we observed balanced decreases in the numbers of multiple subsets within the B lineage without evidence of selectivity among main sub-classes of B cell. Moreover, the surface expression of IgM was comparable in *Dhrs7bΔ/Δ* mice versus WT controls (*Figure 2—figure supplement 1*, panels A-F). Thus, the frequencies of MZB and FOB amidst the smaller population of splenic B cells in tamoxifen-treated *Dhrs7b^f/f; huCD20-CreER^T2* mice were similar (*Figure 2—figure supplement 1*, panel B), and frequencies of B1 B cells in the peritoneal cavity also were unaffected (*Figure 2—figure supplement 1*, panel E). Prior work documents normal frequencies and numbers of developing and mature B cells under harsher conditions of tamoxifen-induced deletion with CreER^T2 (*Higashi et al., 2009*). De novo B cell production rates are too low to yield the ~20% decrease observed 1 week after gene disruption, and few splenic B cells are in cell cycle. Accordingly, we infer that after maturation of a B cell, *Dhrs7b* promotes its survival, albeit to a modest degree.

B cell population growth can be impacted by increased cell death and/or decreased proliferation. To test if PexRAP affects proliferation in vivo, *Dhrs7bΔ/Δ* B cells were stained with CellTrace Violet (CTV) and adoptively transferred into B-cell-deficient μMT recipient mice. Strikingly fewer PexRAP-depleted B cells were recovered (*Figure 2D*), and the frequencies of *Dhrs7bΔ/Δ*B cells that were IgD^neg (i.e. had been activated) were half those of controls (*Figure 2E*; *Figure 2—figure supplement 1*, panel G). As compared to control B cells, CTV partitioning analyses suggested that division rates in vivo were modestly lower for the *Dhrs7bΔ/Δ* B cells (*Figure 2E and F*). Mitogen-stimulated *Dhrs7bΔ/ΔB* cells that proliferated in culture exhibited substantially less robust division: frequencies of viable cells that had undergone ≥3 divisions were halved in *Dhrs7bΔ/Δ* B cells and yielded ~1/3 as large a progeny population (*Figure 2G–I*; *Figure 2—figure supplement 1*, panel H). Collectively, these data indicate that the *Dhrs7b* gene product supports homeostatic maintenance of a quiescent (pre-immune) B cell population in vivo and effective proliferation of B cells.

## B cell expression of PexRAP is required for achieving normal concentrations of ether phospholipids

As noted, lipidomic analyses that used two-dimensional image mass spectrometry (2D-IMS) discovered that local concentrations of a subset of ether lipids are enriched in GC compared with the area outside of GC after immunization (*Jones et al., 2020*). To measure the extent to which PexRAP expression within mature B cells can affect their lipid content and composition, including their ether- and plasmal-ogen phospholipids, LC-MS-MS analyses were performed with B cells directly isolated from spleens as well as those cultured after mitogenic activation. These analyses showed substantial reductions in many ether phospholipids, as well as lysophosphatidylethanolamines (LPE) generated by phospho-lipase cleavage of the R2 fatty acid sidechain of an ether phospholipid, in PexRAP-deficient B cells (*Figure 3A and B*; *Figure 3—figure supplement 1*, panels A, B). While beyond the scope of this work, this evidence suggests that the generation of potential signaling molecules via phospholipase(s) such as PLA2 may be reduced. Of note, *Dhrs7b* inactivation reduced levels of a subset of plasmalogens in

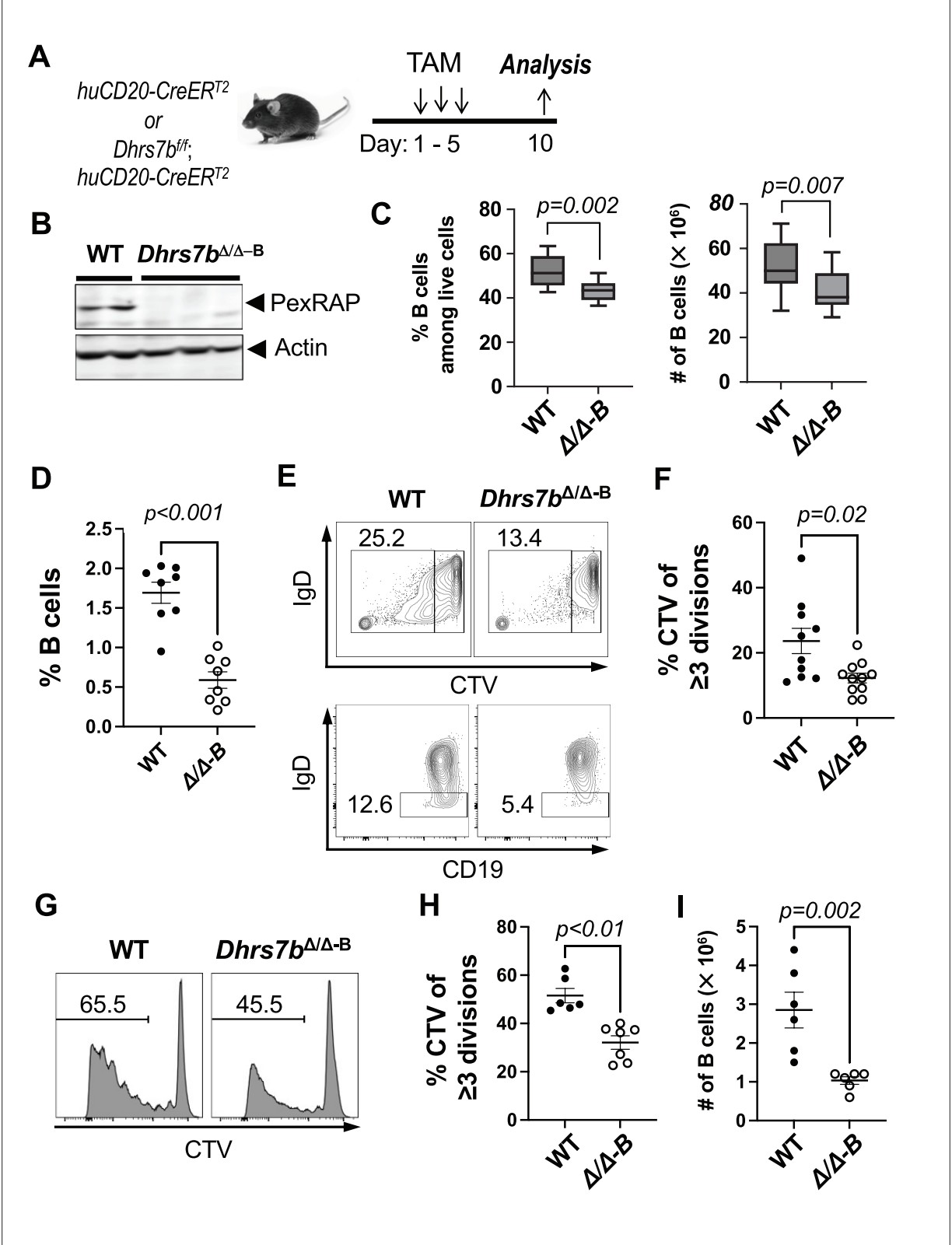

**Figure 2.** PexRAP promotes proliferation of B cells. (**A**) Schematic of *Dhrs7b* inactivation in mature mice via tamoxifen treatment and a B cell type-specific conditional allele. Mice (*huCD20*-CreER^T2; *Dhrs7b*^+/+, or *huCD20*-CreER^T2; *Dhrs7b*^f/f) were injected with tamoxifen (d1, 3, 5) and harvested at day 10. (**B**) Deletion efficiency of conditional *Dhrs7b* alleles. After in vivo tamoxifen injections, B cells were isolated from spleens of *huCD20*-CreER^T2 or *Dhrs7b*^f/f; *huCD20*-CreER^T2 mice, as indicated, and analyzed by immunoblotting as in *Figure 1B*. (**C**) PexRAP and the maintenance of B cells. Shown

*Figure 2 continued on next page*

*Figure 2 continued*

are the frequencies and the numbers of CD19+ B220+ B cells among viable lymphocytes in spleen (left and right panels, respectively). Data are pooled from four independent replicate experiments (n=12 WT and 12 cKO). Shown in box and whisker plots are the means, with whiskers that extend to the minimum and maximum values and boxes that outline upper and lower quartile values with the midline identifying the median. (**D–F**) PexRAP regulates B cell proliferation in vivo. CTV-labeled B cells were adoptively transferred into μMT recipient mice and analyzed 4 days thereafter. Shown are the frequencies of B220+ CD19+ events among splenocytes in the viable cell gate (**D**), along with representative flow plots of CTV partitioning and surface IgD in B cell gates, with rectangles defining the gating for divided <or ≥ 3 x, and IgD+ vs IgDneg. A similar difference was observed in analyzing divided vs undivided (**E**). (**F**) Aggregated frequencies of divided B cells from five independent replicate experiments, as defined in (**E**) (n=10 WT and 11 cKO). Additional data on the IgDneg population are in ***Figure 2—figure supplement 1G***. (**G–I**) PexRAP promotes B cell proliferation in vitro. After in vivo tamoxifen injections, bead-purified B cells from spleens of CreERT2 mice (WT and *Dhrs7bΔ/Δ, i.e. cKO*) were stained with Cell Trace Violet (CTV), activated and cultured for 4 days in anti-CD40, BAFF, IL-4, IL-5, and 4-hydroxytamoxifen in three biologically independent experiments totaling 6 WT and 7 cKO mice. (**G**) Representative flow-cytometric analysis of CTV partitioning, with the gating line denoting multiply divided cells, with inset numbers representing the frequencies of such B cells in each of the two plots shown. (**H**) Quantified frequencies of divided ≥3 times. (**I**) Numbers of B cells recovered at the end of the cultures. Additional information is in ***Figure 2—figure supplement 1***.

The online version of this article includes the following source data and figure supplement(s) for figure 2:

**Source data 1.** Raw images for the immunoblots shown in ***Figure 2B***.

**Source data 2.** Uncropped images for the immunoblots shown in ***Figure 2B***, with relevant bands labelled.

**Figure supplement 1.** PexRAP is dispensable for pre-immune B cell subset balance and surface IgM expression, but contributes to B cell population growth.

naive B cells (***Figure 3A, B and D***; ***Figure 3—figure supplement 1***, panel A), although changes of others were modest or absent in the naive population. A greater impact of PexRAP on quantitative amounts of phospholipids was found in the activated B cells. Several species, skewed towards those with the most detected ions, a number of which were at less than 1/5th the level of non-deleted control B cells (***Figure 3C and D***). Phospholipid results observed with naive and activated B cells from tamoxifen-treated CreERT2-expressing mice were similar to those of unmanipulated controls (***Figure 3—figure supplement 1***, panels C, D). There is no in vitro surrogate that faithfully represents GC or the GC B cell, but collectively these data provide strong evidence that the capacity to rapidly produce increased levels of many ether lipid molecules after lymphocyte activation depends on a functional *Dhrs7b* gene in B cells. The data provide evidence that *Dhrs7b* in B cells enhances cell-autonomous biosynthesis of some plasmalogens and other ether lipid species. Especially in resting B cells that are out of cycle, additional uptake or biosynthesis pathways may add to the overall pool (***Lodhi et al., 2015b***; ***Lodhi et al., 2012***). Notwithstanding these potential contributions, however, the straightforward inference is that PexRAP catalyzes the generation of ether lipid precursors in amounts crucial for regulating the overall ether phospholipid pools, especially after B cell activation.

Our earlier work (***Jones et al., 2020***) reported only ion mode features in negative mode and did not analyze the primary follicle (the vast majority of which are resting B cells) relative to the extrafollicular white pulp, red pulp, and GC. To investigate the requirement for specific expression of PexRAP, we started with AID-GFP mice to enhance spatial localization. These measurements confirmed that a substantial number of both positive and negatively charged ions whose exact masses identified them as plasmalogens were substantially concentrated in GC as compared to the rest of the B cell follicle (***Table 1***; ***Figure 4—figure supplement 1***, panels A, B). Accordingly, we tested if the differential enrichment of any ether lipid species in primary or secondary follicles depends on biosynthesis within a specific lymphoid lineage. Mice with conditional mutations were tamoxifen-treated, immunized with SRBC, harvested seven days thereafter, and analyzed by IMS (***Figure 4A***). With deletion driven by the widely expressed *Rosa26-CreERT2* transgene, as in earlier work ***Lodhi et al., 2015b*** and ***Figure 1***, substantial reductions of the GC enrichment pattern were observed (***Figure 4B***). For instance, two representative ions identified in negative ion mode (i.e. *m/z* 752.5545 and *m/z* 776.5556) were again enriched in GC regions of spleens from immunized WT mice. These features barely increased in splenic samples of immunized *Dhrs7bΔ/Δ* (tamoxifen-injected *Dhrs7bf/f; Rosa26-CreERT2*) mice (***Figure 4B***).

To determine if the levels of particular ether phospholipids in GC regions were affected by PexRAP expression in B cells, *Dhrs7bf/f; huCD20*-CreERT2 mice and *huCD20*-CreERT2 controls were analyzed after tamoxifen injections followed by immunization and compared to samples from unimmunized mice (***Figure 4C***). IMS using both positive and negative ion modes identified at least eight ions - including *m/z* 752.5545, *m/z* 776.5556, and *m/z* 872.5749 - much more substantially (~two- to

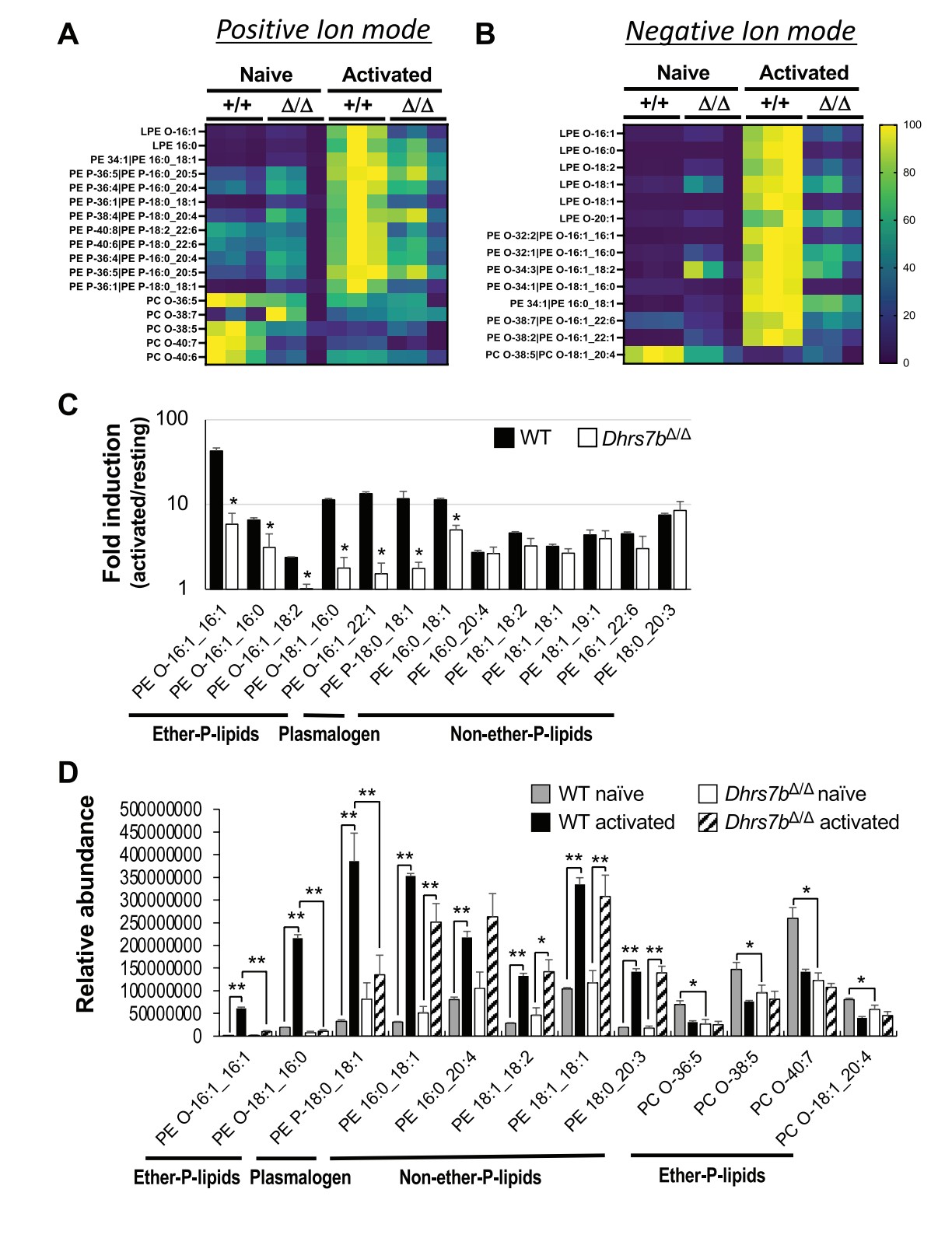

**Figure 3.** B cell expression of PexRAP is essential for normal concentrations of many ether phospholipids. Splenic B cells (CreER^T2+ or CreER^T2+, *Dhrs7b* Δ/Δ) from tamoxifen-injected mice were analyzed by LC-MS-MS either directly ex vivo ('Naïve') or after activation and culture (48 h) ("Activated") as in *Figure 2*. Shown are z-scored heat maps for the relative abundance of subsets of ether and plasmalogen phospholipids and lysophospholipids identified by exact mass and secondary fragmentation in (**A**) positive and/or (**B**) negative ion modes. PE, phosphatidylethanolamine; PC,

*Figure 3 continued on next page*

*Figure 3 continued*

phosphatidylcholine; LPE, lysophosphatidylethanolamine. More species are displayed in *Figure 3—figure supplement 1*. (**C**) Quantitated peak areas for activation-induced increases ('fold-induction' ratios of activated/resting, plotted on $\log_{10}$ scale) for the indicated ether, plasmalogen, and diacyl (non-ether) phospholipids in the PexRAP-sufficient and -depleted B cells. * denotes species for which -<0.05 for the effect of *Dhrs7b* inactivation. (**D**) Shown are the raw values of mean peak areas for the indicated species in the freshly purified and the activated B cells of the indicated genotypes, as indicated. (Three independent samples from individual mice of each genotype were analyzed.) * $p<0.05$, ** $p<0.01$ by unpaired Student's t-test. Additional data, including controls comparing B cells of unmanipulated B6 mice to those expressing CreER[T2] and injected with tamoxifen, are in *Figure 3—figure supplement 1*.

The online version of this article includes the following figure supplement(s) for figure 3:

**Figure supplement 1.** Activation of B cells induces PexRAP-dependent increases in their ether and plasmalogen P-lipids.

threefold) concentrated in GC of immunized control mice (*Figure 4D and E*; *Table 2*) than in the primary follicle of unimmunized mice. Notably, these increased signals in secondary follicles (GC) were reduced or almost completely eliminated in GCs of mice immunized after B-cell-specific PexRAP depletion (*Dhrs7b* ΔB cells; *Figure 4D and E*; *Table 2*; *Figure 4—figure supplement 1*, panel C). Collectively, these results indicate that *Dhrs7b* gene expression in activated B cells regulates the spectrum of ether lipids in the micro-anatomic locale of a secondary follicle. Interestingly, immunization also led to modest increases in the levels of some ether and conventional phospholipids in the B-cell-rich primary follicle regions (*Table 2*). This finding extends the previous report (*Jones et al., 2020*), which focused on the relationship between the AID-GFP[hi] region (i.e. GC) and lipid features. Moreover, the absence of PexRAP blunted most of the immunization-induced increases observed in the primary follicles. These findings and those of *Figure 3* suggest that there are secondary consequences of *Dhrs7b* inactivation (e.g. some di-acyl phospholipids are affected), but provide direct evidence that the enhancement of many ether phospholipid signals in the GC depends on PexRAP in B cells.

**Table 1.** Immunization-induced increases of selected ether lipids in GC.

Shown are (a) a sample of *m/z* features characteristic of the indicated phospholipid species identified by the LIPIDMAPS database, and (b) the mean (± SEM) ion intensity / counts in IMS data generated from spleens of AID-GFP mice, immunized or not (UI), after mapping to the indicated regions of interest (primary follicles or GC, i.e. secondary follicles) using fluorescent images. Shown are ions more accumulated in GC regions compared to primary follicles: 8 ions from negative ion mode and 5 ions from positive ion mode, all $p<0.05$ for comparison of 1[0] follicle to UI (c), or GC to 1[0] follicle (d). p Values were calculated by Mann-Whitney U test.

**Negative ions**

| m/z [a] Lipid ID | UI | 1° B cell follicle | GC |
|---|---|---|---|
| 716.5216 PE(18:0_16:1) | 618±22 [b] | 2400±83 [c] | 3674±154 [d] |
| 752.5545 PE(O-18:0_20:4) | 301±14 | 1739±33 [c] | 1960±52 [d] |
| 760.5056 PE(16:0_22:6) | 203±4 | 1816±30 [c] | 2031±23 [d] |
| 772.5242 PE(P-18:1_22:6) | 234±5 | 2334±47 [c] | 2578 ± 63 [d] |
| 776.5556 PE(O-18:0_22:6) | 277±9 | 1638±51 [c] | 2074±90 [d] |
| 869.5526 PI(O-18:0_20:5) | 242±7 | 1611±24 [c] | 1789±15 [d] |
| 872.576 PC(18:0_24:0) | 171±6 | 1167±16 [c] | 1369±18 [d] |
| 888.5612 PS(22:1_22:6) | 506±13 | 2684±54 [c] | 3386±72 [d] |

**Positive ions**

| m/z [a] Lipid ID | UI | 1° B cell follicle | GC |
|---|---|---|---|
| 739.4587 PG(12:0_20:3) | 3042±102 | 3733±116 [c] | 4778±274 [d] |
| 740.465 PC(16:0_15:1) | 1310±13 | 1478±66 [c] | 1933±142 [d] |
| 784.5564 PC(10:0_24:0) | 2840±24 | 7956±147 [c] | 9856±270 [d] |
| 798.5347 PC(10:0_25:0) | 9937±269 | 27589±574 [c] | 34733±1,644 [d] |
| 799.5375 PG(18:0_18:1) | 5665±65 | 12078±282 [c] | 15311±790 [d] |

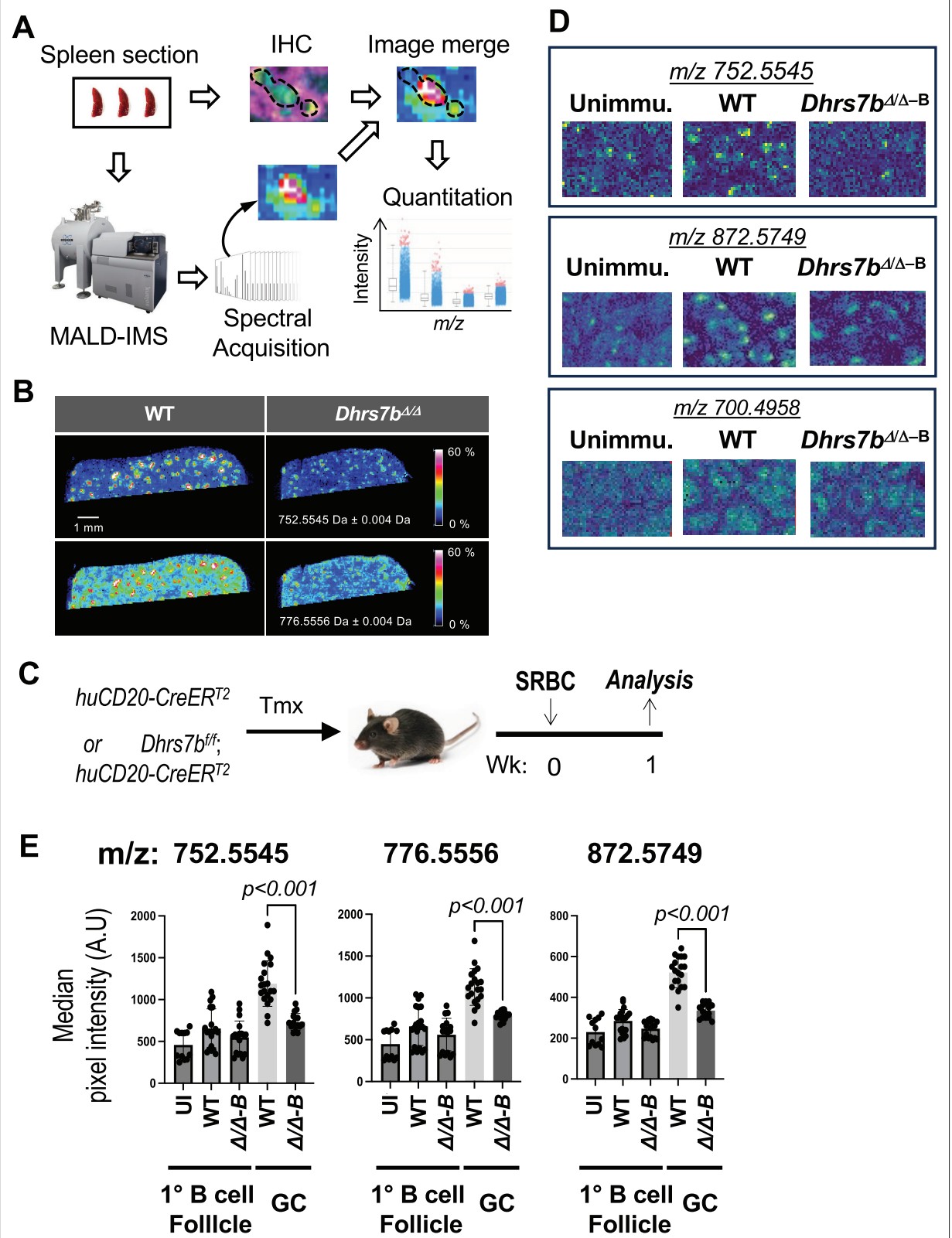

**Figure 4.** PexRAP is essential for normal concentrations and distributions of some ether phospholipids in splenic follicles. (**A**) Schematic diagram illustrating the 2D-IMS analysis work flow, image merging and quantitation. Spleens harvested 1 week after immunization with SRBC were used to generate serial tissue sections followed by immunofluorescence (IF) staining of one section and IMS analysis with the adjacent one. IF and IMS images were aligned to map ion intensity distributions to microanatomic regions (B cell follicle and GC). The intensities of specific ions on B cell follicles and GC

*Figure 4 continued on next page*

*Figure 4 continued*

regions were quantitated as described in *Materials and methods*. (**B**) Identification of ether lipid species localizing to lymphoid follicles. Representative ion images of two ions [*m/z* 752.5545, and *m/z* 776.5556] with spleens from immunized mice (WT and *Dhrs7bΔ/Δ*) as shown in **Figure 1A**. (**C**) Schematic of immunization for IMS analyses of B-cell-specific PexRAP loss. Mice of the indicated genotypes (huCD20-CreER$^{T2}$±*Dhrs7 b$^{f/f}$*) were treated with tamoxifen, immunized with SRBC, and harvested 1 week after immunization. (**D**) Identification of ether lipid species localizing to lymphoid follicles. Representative ion images of three ions [*m/z* 752.5545, *m/z* 872.5749, and *m/z* 700.4958] in mass spectrometry imaging of spleens from WT and *Dhrs7b* Δ B mice (shown as *Dhrs7bΔ/Δ−$^B$* samples). Immunofluorescent images at higher magnification, delineating LZ and DZ marked by CD35 staining, along with quantification of sizes of GC and their LZ and DZ, are shown in **Figure 5—figure supplement 1A,B**. (**E**) Shown in the bar graphs are the mean (± SEM) ion intensities in primary lymphoid follicles (B cell zones) and GC from spleens of WT and *Dhrs7b* Δ B mice, immunized or not ('UI') as indicated. Median intensity of each ion was obtained from 3 follicles / spleen and 3 GC / spleen from WT and *Dhrs7b* Δ B mice (three biological replications comprising 7 WT and 6 cKO spleens). p values were calculated by Mann-Whitney U test. Additional data are in **Figure 4—figure supplement 1**.

The online version of this article includes the following figure supplement(s) for figure 4:

**Figure supplement 1.** Immunization induces increases in P-lipids of both primary and secondary follicles dependent on PexRAP in B cells.

## B-cell-intrinsic role of *Dhrs7b* in Ab affinity and quantity

As key components of adaptive immunity, progeny of an activated B cell can differentiate into Ab-secreting plasma cells (PC) or into GC B cells. Over the course of an immune response, the GC reaction diversifies the affinity for and breadth of antigen recognized by the original BCR and improves properties of memory (*Good-Jacobson and Shlomchik, 2010*; *De Silva and Klein, 2015*; *Cyster and Allen,*

**Table 2.** Impact of PexRAP on relative quantities of selected lipid species in primary and secondary follicles (GC).

As in **Table 1** except B cells of immunized mice were either WT or lacked PexRAP (cKO, i.e., *Dhrs7b* Δ B). Shown are (a) a sample of *m/z* features characteristic of the indicated phospholipid species. (b) the mean ± SEM ion intensity counts in IMS data after mapping to the indicated regions of interest as in **Table 1**. (c-f) $p<0.05$ for the null hypothesis in considering a difference between spleens of immunized WT mice versus UI controls (c), 1$^0$ B cell follicles in spleens of immunized WT vs *Dhrs7b* Δ B mice (d), GC vs in 1$^0$ B cell follicles in spleens of immunized WT mice (e), and GC of immunized WT vs *Dhrs7b* Δ B mice (f) in the ion counts for designated *m/z* features. p values were calculated by Mann-Whitney U test.

**Negative ions**

| m/z [a] Lipid ID | UI | WT follicle | PexRap cKO follicle | WT GC | PexRap cKO GC |
|---|---|---|---|---|---|
| 436.2801 LPE (O-16:1) | 315±4 | 369±7 | 374±11 | 610±20 [e] | 475±19 |
| 716.5216 PE (18:0_16:1) | 754±43 [b] | 1608±182 [c] | 1263±112 | 2729±247 [e] | 1770±137 [f] |
| 746.5098 PE (O-16:1_22:6) | 622±19 | 672±35 | 609±23 | 1066±39 [e] | 836±39 |
| 752.5545 PE (O-18:0_20:4) | 459±49 | 657±51 [c] | 548±46 [d] | 1191±61 [e] | 734±25 [f] |
| 760.5056 PE (16:0_22:6) | 229±8 | 386±25 [c] | 312±16 [d] | 569±27 [e] | 408±15 [f] |
| 772.5242 PE (P-18:1_22:6) | 388±47 | 617±46 [c] | 499±33 [d] | 918±21 [e] | 646±9 [f] |
| 776.5556 PE (O-18:0_22:6) | 449±53 | 661±50 [c] | 562±46 | 1130±49 [e] | 789±13 [f] |
| 869.5526 PI (O-18:0_20:5) | 509±81 | 641±50 [c] | 494±40 [d] | 1014±21 [e] | 690±12 [f] |
| 872.576 PC (18:0_24:0) | 230±19 | 285±12 [c] | 247±9 [d] | 522±17 [e] | 336±9 [f] |
| 888.5612 PS (22:1_22:6) | 1423±277 | 2051±199 [c] | 1904±199 | 3260±156 [e] | 2629±80 [f] |

**Positive ions**

| m/z [a] Lipid ID | UI | WT follicle | PexRap cKO follicle | WT GC | PexRap cKO GC |
|---|---|---|---|---|---|
| 730.5767 PE (P-18:0_18:1) | 91±3 | 97±3 | 85±3 | 130±7 [e] | 102±4 |
| 739.4587 PG (12:0_20:3) | 3042±102 | 3094±40 | 2944±31 [d] | 3506±140 [e] | 3011±46 [f] |
| 784.5564 PC (10:0_24:0) | 2840±24 | 2917±74 | 2617±46 [d] | 3522±102 [e] | 2561±42 [f] |
| 798.5347 PC (10:0_25:0) | 9937±269 | 10928 ± 214 [c] | 11372±155 [d] | 15378±116 [e] | 14694±186 [f] |
| 801.5509 PG (18:0_18:0) | 1540±37 | 1591±46 | 1633±26 | 1444±27 [e] | 1533±22 [f] |
| 847.5361 PI (15:0_18:0) | 1277±24 | 1339±54 | 1217±35 | 1150±46 [e] | 1044±44 |
| 874.5623 PC (20:5_22:6) | 754±39 | 688±48 | 675±25 | 469±27 [e] | 556±37 |

*2019*; *Victora and Nussenzweig, 2022*; *Inoue and Kurosaki, 2024*). PCs that develop after a second encounter with antigen increase the amounts of high-affinity Ab. The higher quality and quantity of Ag-specific Abs from plasma cells are integral to humoral immune responses (*Inoue and Kurosaki, 2024*). An increasing body of evidence indicates that metabolic reprogramming and intermediary metabolites modulate immune cell differentiation and function (*Chapman and Chi, 2022*; *Bacigalupa et al., 2024*; *Jellusova, 2020*; *Boothby et al., 2022*; *O'Neill et al., 2016*; *Cho et al., 2016*; *Weisel et al., 2020*; *Chen et al., 2021*; *Urbanczyk et al., 2022*; *Yazicioglu et al., 2023*; *Brookens et al., 2024*). The observed B-cell-intrinsic function of *Dhrs7b* in the accumulation of many ether lipid species in both primary and secondary follicles prompted us to test the effect of B cell type-restricted PexRAP depletion on GC responses and Ag-specific Ab production. Tamoxifen-injected *Dhrs7b^{f/f}; huCD20-CreER^{T2}* and control mice were immunized with NP-ovalbumin (NP-OVA), then boosted with NP-OVA to elicit affinity-matured Ab and harvested 1 week thereafter (*Figure 5A*). Although PexRAP-deficient B cell numbers were reduced by less than 20% a week after starting gene inactivation prior to immunization (i.e. were over 0.8-fold those of controls; *Figure 2A–C*), frequencies and numbers of IgD^{neg} B cells were halved when mice with B-cell-specific inactivation of *Dhrs7b* were harvested (4 weeks after completion of the induced deletion; *Figure 5B and C*). Moreover, the GL7^+ CD95^+ fraction of the B cell population in the dump / IgD^{neg} gate was substantially reduced (*Figure 5D*), such that numbers of GL7^+ CD95^+ GC B cells were dramatically decreased in *Dhrs7b* ΔB mice (*Figure 5E*). Consistent with these findings, GC in *Dhrs7b* ΔB mice was reduced after SRBC immunization (*Figure 5F and G*; *Figure 5—figure supplement 1*, panels A, B), and the numbers of centrocytes and centroblasts were dramatically lower in *Dhrs7b* ΔB mice compared with WT controls (*Figure 5H*). LZ B cells (CD86^+ CXCR4^{lo}) trended toward being at lower prevalence in *Dhrs7b* ΔB mice, whereas their DZ counterparts (CD86^{neg} CXCR4^+) were comparable to wild-type controls, resulting in a modest decrease in the LZ/DZ ratio when B cells lacked PexRAP (*Figure 5—figure supplement 1*, panel C). The frequencies of CD138^+ GL7^+ IgD^- early plasmablasts were comparable between *Dhrs7b* ΔB mice and control mice (*Figure 5—figure supplement 1*, panel D). Although the loss of function was B cell specific, the frequencies of Tfh and GC-Tfh cells were decreased in *Dhrs7b* ΔB mice compared with control mice (*Figure 5—figure supplement 1*, panel E). We infer that support for the Tfh populations was reduced due to impact(s) on GC B cell numbers and/or function.

Ab class-switch recombination (CSR) and affinity maturation can occur through extrafollicular responses. However, the GC reaction significantly increases Ab diversification and high-affinity Ab production, in part later in a primary response but also upon secondary exposure to antigens (*MacLennan et al., 2003*; *Rajewsky, 1996*; *Boothby et al., 2019*; *De Silva and Klein, 2015*; *Cyster and Allen, 2019*; *Victora and Nussenzweig, 2022*). ELISA performed with sera at 3 week post-immunization (*Figure 6A–D*), just before the boost, showed that B-cell-specific depletion of PexRAP prior to immunization reduced NP-specific IgM and the switched isotype IgG1 (*Figure 6A and C*). Of note, the capacity to generate high-affinity Ab, detected with low valency NP_2, was even more severely undermined than the overall response - especially for IgG1 (*Figure 6B and D*). A week after a second immunization - an interval similar to that which followed immunization with SRBC - Ab-secreting cells (ASCs) in the spleen (*Figure 6E*) and circulating anti-NP IgM concentrations were diminished in *Dhrs7b* ΔB mice compared with controls (*huCD20*-CreER^{T2}, *Dhrs7b^{+/+}* treated with tamoxifen in parallel to the mice with induced loss of PexRAP) (*Figure 6F*). The ratio of high-affinity (NP_2-binding) to all-affinity (NP_{20}) IgM Ab also was substantially lower in *Dhrs7b* ΔB mice (*Figure 6G*), as were the serum levels of high-affinity IgG1 and IgG2c, class-switched isotypes (*Figure 5—figure supplement 1*, panels F, G). These data reinforce and extend the conclusion that the expression of PexRAP in mature B cells is important for their physiology and function.

Consistent with in vivo results finding reduced Ag-specific ASCs in *Dhrs7b* ΔB mice compared with controls (*Figure 6E*), the attenuated population of PexRAP-deficient B lineage cells recovered after transfer into recipient mice (*Figure 2D*) yielded lower frequencies of CD138^+ progeny (*Figure 6H and I*) along with evidence of reduced activation (i.e., lower frequencies of IgD^- progeny [*Figure 2—figure supplement 1*], panel G). Mitogen-stimulated *Dhrs7bΔ/ΔB* cells also yielded lower frequencies of CD138^+ cells in vitro, but the division-specific frequencies of CD138^+ cells showed only modest decreases in *Dhrs7bΔ/ΔB* cells (*Figure 6—figure supplement 1*, panels A, B). Thus, the reduction of CD138^+ cell differentiation appears to be due mostly to its dependence on survival to a sufficient division count.

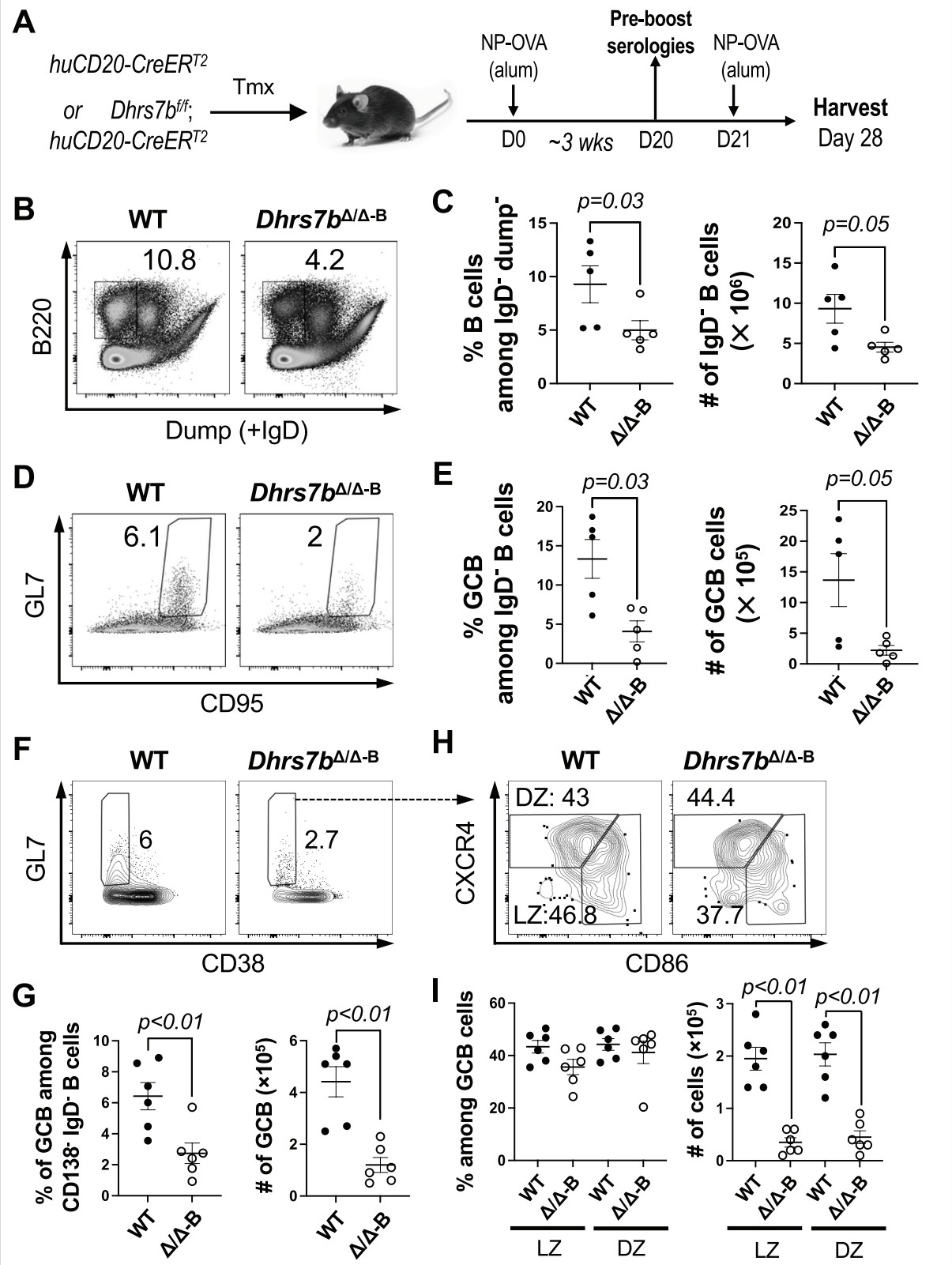

**Figure 5.** B-cell-intrinsic role of PexRAP in GC response. (**A**) Schematic of immunization with NP-OVA in alum after inactivation of *Dhrs7b* in B lineage cells. Tamoxifen-treated WT (*huCD20*-CreER[T2]; *Dhrs7b*[+/+]) or *Dhrs7b* Δ B mice (*huCD20*-CreER[T2]; *Dhrs7b*[f/f]) were immunized with NP-OVA, with sera collected 3 weeks thereafter ('1⁰ response'), followed by boosting with NP-OVA and harvest 1 week after the 2nd immunization. (**B**, **C**) Representative flow plots of splenic IgD[neg] B cells (**B**) at the time of harvest (1 week after 2nd immunization), and aggregate data (**C**) for two replicate experiments

*Figure 5 continued on next page*

*Figure 5 continued*

(n=5 WT and 5 cKO). The cocktail of reagents for the dump channel included anti-IgD. (**D**, **E**) B-cell-intrinsic function of PexRAP in GC B cell response. Representative flow plots of GL7$^+$ CD95$^+$ GC B cells among viable IgD$^{neg}$ B cells (**D**), and aggregated frequencies and numbers of GC B cells for two replicate experiments (**E**). Mann-Whitney U test was used to calculate p values. (**F**, **G**) WT (*huCD20*-CreER$^{T2}$; *Dhrs7b$^{+/+}$*) or *Dhrs7b* Δ B mice (*huCD20*-CreER$^{T2}$; *Dhrs7b$^{f/f}$*) were injected with tamoxifen and immunized with SRBC as in *Figure 4A*. Shown are the representative flow plots of GL7$^+$ CD38$^-$ GC B cells among IgD$^{neg}$ CD138$^{neg}$ viable B cells (**F**), and aggregate data (**G**) for two replicate experiments (n=6 WT and 6 cKO). (**H**) PexRAP is mostly dispensable for the balance of LZ and DZ B cells. The graph shows the mean (± SEM) frequencies (left) and the numbers (right) of CD86$^+$ CXCR4$^{lo}$ LZ B cells and CD86$^{neg}$ CXCR4$^+$ DZ B cells among IgD$^{neg}$ CD38$^{neg}$ CD138$^{neg}$ GL7$^+$ GC cells. Mann-Whitney U test was used to calculate p values. Additional data are in *Figure 5—figure supplement 1*.

The online version of this article includes the following figure supplement(s) for figure 5:

**Figure supplement 1.** A subset of ether phospholipids in primary follicles and GC (secondary follicles) depends on PexRAP in B cells.

*Dhrs7b* inactivation was initiated prior to immunization in the preceding experiments. Therefore, the impact of PexRAP deficiency on GC and the Ab response in such a setting might be due exclusively to impairment of B cells, for example their reduced population expansion, prior to their entry into GC. Alternatively, the effects could also in part involve a requirement for PexRAP-dependent metabolites within GC B cells. To test if *Dhrs7b* functions within GC, we used conditional deletion of *Dhrs7b* driven by the *S1pr2-CreER$^{T2}$* transgene whose expression at high levels marks GC B cells (*Shinnakasu et al., 2016*). Of note, experiments with a fate-marking reporter allele showed that activated B cells that lack the GC B phenotype were not marked by this conditional Cre after immunization with the NP-carrier approach (*Shinnakasu et al., 2016*). *Dhrs7b$^{f/f}$*; *S1pr2-CreER$^{T2}$* and control *S1pr2-CreER$^{T2}$* (control) mice were immunized with SRBC, and tamoxifen was injected at a time point after the initiation of GC to test more directly that the inactivation of *Dhrs7b* would be in GC B cells rather than pre-GC blasts (*Figure 7A*). Frequencies of GC B cells were substantially lower in tamoxifen-treated *Dhrs7b$^{f/f}$*; *S1pr2-CreER$^{T2}$* mice (*Figure 7B and C*). While this finding does not exclude that there may be an additional effect of impaired clonal expansion in B cells activated by immunization but not yet resident in the GC, the results indicate that PexRAP functions within GC B cells to support a full population of this subset. All together, these data provide evidence that *Dhrs7b* expression influences the lipidome and function of GC B cells. Notably, an optimal GC response requires PexRAP function in B cells, and the B-cell-intrinsic functions of *Dhrs7b* promote the quantity and affinity maturation of Ab responses, in part via net proliferation of B cells.

### *Dhrs7b* contributes to modulation of ROS and their impact on B cell population growth

Increased susceptibility to cell death after activation would be a mechanism that could cause the reduced population expansion, GC, and Ab production. Consistent with earlier analyses of LPS-stimulated B cells (*Price et al., 2018*), we had noted that ROS levels were higher in GC B cells, memory B cells (MBCs) and ASCs than in the naive B2 B cell pool (*Brookens et al., 2024*). ROS can be produced in activated B cells via diverse sources that include mitochondrial electron transport chain function, NADPH oxidase activated by BCR engagement, and fatty acid oxidation in peroxisomes (*Capasso et al., 2010*; *Wheeler and Defranco, 2012*; *Tsubata, 2020*). ROS can positively mediate signal transduction, but their steady-state concentrations need to be calibrated because overproduction can have deleterious effects on cells, for instance via lipid peroxidation at excessive rates (*Jiang et al., 2021*; *Nordgren and Fransen, 2014*; *Liang et al., 2022*). Endogenous antioxidant properties have been imputed to plasmalogens because the vinyl ether bond is susceptible to reaction with reactive oxygen, thereby scavenging ROS (*Dean and Lodhi, 2018*; *Braverman and Moser, 2012*; *Gorgas et al., 2006*; *Jiménez-Rojo and Riezman, 2019*), so we tested if *Dhrs7b* function contributes to redox control in B cells. ROS levels were ~twofold higher in GC B cells from immunized *Dhrs7b* ΔB mice compared to those from immunized WT controls (*Figure 7D*). *Dhrs7b*Δ/Δ GC B cells also showed higher cell death and attenuated BrdU incorporation (*Figure 7E and F*; *Figure 6—figure supplement 1*, panel C). Moreover, CD40-stimulated *Dhrs7b*Δ/ΔB cells exhibited ~threefold higher signal and about a doubling when stained with fluorescent sensors of cellular and mitochondrial ROS, respectively, compared to WT B cells in vitro (*Figure 8A–D*). Substantially increased ROS were also measured after activation by BCR and CD40 cross-linking (*Figure 6—figure supplement 1*, panel D). Iron- and copper-dependent lipid peroxidation is considered to be a biological mechanism for ROS-mediated

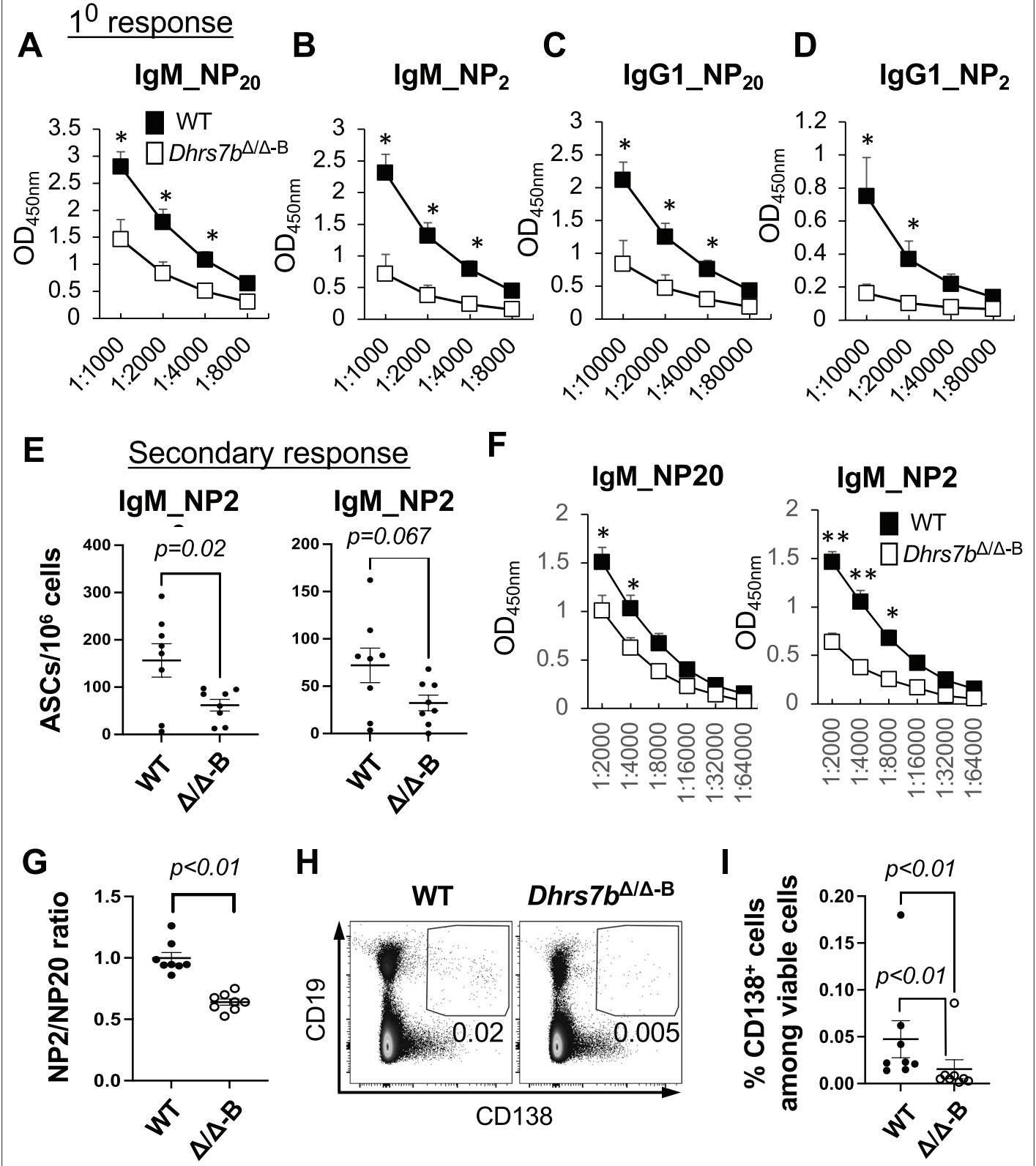

**Figure 6.** Ab response and affinity increase promoted by PexRAP in B cells. Tamoxifen-treated mice (*huCD20*-CreER[T2]; *Dhrs7b*[+/+] or *huCD20*-CreER[T2]; *Dhrs7b*[f/f], i.e. *Dhrs7b* Δ B; designated *Dhrs7bΔ/Δ−B*) were immunized as in *Figure 5*, with venous blood collected to measure Ag-specific Ab in the 1⁰ response just prior to a second immunization, followed by harvesting a week thereafter (2⁰ response). (**A–D**) Ag-specific Ab in primary response sera, prior to the boost. Shown are the all- ($NP_{20}$) and high- ($NP_2$) -affinity anti-NP IgM (**A**, **B**) and IgG1 (**C**, **D**), as indicated, with $NP_{20}$ a high hapten density

*Figure 6 continued on next page*

*Figure 6 continued*

to detect both low- and high-affinity Ab and $NP_2$ a low hapten density selective for high-affinity Ab. (**E**, **F**) Levels of anti-NP IgM detected using $NP_{20}$ and $NP_2$ for ELISpot (**E**) and ELISA (**F**) as described in *Materials and Methods*. Graphs show mean (± SEM) number of Ab-secreting cells in spleen (**E**) and (**F**) mean (± SEM) $OD_{450nm}$ in measurement of NP-specific IgM in serial dilutions of sera. p values were calculated by Student' t-test. * indicates p<0.05, and ** indicates p<0.01. (**G**) PexRAP in B cells promotes Ab affinity maturation. The bar graph shows the mean (± SEM) ratios of high-affinity to all-affinity NP-specific IgM Ab in sera of individual mice (each dot representing one subject) using $OD_{450nm}$ values at the 1:2000 dilution, with data from three independent experiments comprising eight mice of each type (WT; *Dhrs7b* Δ B). p values were calculated by Mann-Whitney U test. (**H**, **I**) PexRAP promotes CD138⁺ cell differentiation in vivo. B cells were adoptively transferred into μMT-recipient mice and analyzed at 4 days after transfer. Representative flow plot of CD138 and CD19 expression (**H**) and aggregated frequencies of CD138⁺ CD19⁺ cells in the viable cell gate (**I**) from four independent replicate experiments. To test for a potential distortion arising from outlier values, statistical testing was performed both with and without their inclusion. Additional data are in *Figure 6—figure supplement 1* .

The online version of this article includes the following figure supplement(s) for figure 6:

**Figure supplement 1.** PexRAP contributes to GC B cell proliferation, ROS homeostasis and B cell population growth.

cell death (*Nordgren and Fransen, 2014*; *Liang et al., 2022*). Lipid peroxidation measured by a fluorescent indicator (C11-Bodipy) in vitro was higher in *Dhrs7bΔ/ΔB* cells compared with WT controls after each type of mitogen activation (*Figure 8E*; *Figure 6—figure supplement 1*, panel E). Chemical elicitation of increased ROS and lipid peroxidation in B cells, independent from B cell activation or proliferation, rapidly led to reduced cell viability and B cell numbers (*Figure 8F*; *Figure 8—figure supplement 1*, panel A). Furthermore, in vitro analyses indicated that B cells lacking PexRAP also exhibited increases in the activated executioner caspase, cleaved caspase-3 (CC3), along with early apoptotic cells that are annexin V⁺ but exclude 7-aminoactinomycin D (*Figure 8G and H*). These data indicate that *Dhrs7b* function supports not only protection against cell death, but also control of cellular and mitochondrial ROS levels and their impact on lipid modification (*Figure 8—figure supplement 1*). Of note, while $H_2O_2$ did not increase this early apoptotic population (*Figure 8—figure supplement 1*, panel C), increasing ROS with menadione drove B cell death preceded by annexin V⁺ state without an increase in C11-Bodipy signal (*Figure 8—figure supplement 1*, panels D-G). We infer that rather than a single mode of death, increased ROS resulting from PexRAP depletion probably stimulates at least two distinct modes of B cell death.

Consistent with the increased steady-state ROS and death, activated *Dhrs7bΔ/ΔB* cells exhibited a defect in B cell population growth in vivo and in vitro (*Figure 2*; *Figure 8A–H*). The well-established ROS scavenger, N-acetyl-L-cysteine (NAC), exerted a concentration-dependent effect that improved the survival and population growth of PexRAP-deficient B cells (*Figure 8I*). We exploited the sub-maximal rescue by a lower concentration of NAC (1 mM) to explore if peroxisomal oxidative metabolism might contribute to the toxicity observed when B cells are PexRAP-depleted. Although thioridazine on its own provided no increase in the B cell population growth, its combination with 1 mM NAC treatment improved growth of the *Dhrs7bΔ/ΔB* cell population compared to either agent on its own (*Figure 8J*). Analyses of B cells activated by BCR crosslinking along with anti-CD40 also found that PexRAP expression in B cells supports their population increase (*Figure 6—figure supplement 1*, panel F). Moreover, the inclusion of NAC in the cultures partially mitigated the inactivation of *Dhrs7b* (*Figure 6—figure supplement 1*, panel F). Of note, measurements of CTV partitioning identified a modest but statistically significant decrease in division counts of PexRAP-deficient B cells under these conditions and found that NAC did not increase this component of proliferation (*Figure 6—figure supplement 1*, panels G, H). These data support the conclusion that sufficient generation of PexRAP-dependent products is crucial for B cell survival. Moreover, this enzyme contributes - directly or indirectly - to a resolution or detoxification of ROS that is critical for population growth of activated B cells.

### *Dhrs7b* in B cells affects their oxidative metabolism and ER mass

Our finding that mitochondrial ROS were increased prompted us to investigate how PexRAP affects physiological functions by measuring respiration in activated B cells. Mitochondrial stress tests with in vitro activated B lymphoblasts measured small but definite decreases in both basal and maximal respiration (oxygen consumption rates, or OCR; *Figure 9A–C*). The altered performance of mitochondria was associated with reductions in calculated ATP generation and proton leak as well as a small decrease in spare respiratory capacity (SRC) (*Figure 9D–F*, respectively). In contrast, the rates of glucose-stimulated extracellular acidification - cytosolic reactions and a surrogate that can

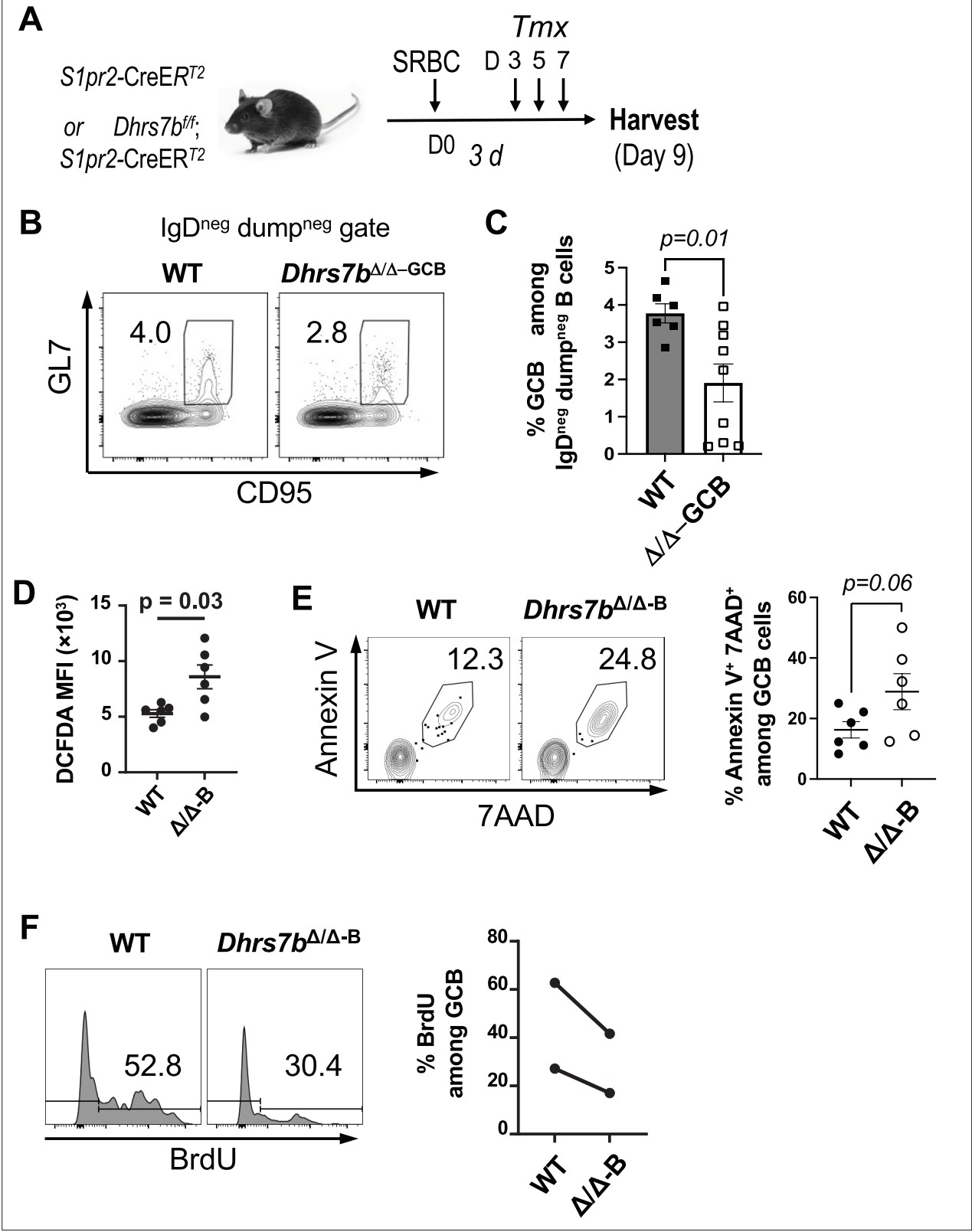

**Figure 7.** Function of PexRAP in GC response. (**A**) Schematic of the time line, with immunization followed later by tamoxifen injections into *S1pr2*-CreER^T2 mice (*Dhrs7b*^+/+ or *Dhrs7b*^f/f). Note that deletion is only initiated just as the germinal center reaction starts (~3.5 day post-immunization). Mice (*S1pr2*-CreER^T2; *Dhrs7b*^+/+ or *S1pr2*-CreER^T2; *Dhrs7b*^f/f) were immunized with NP-OVA, treated with tamoxifen on days 3, 5, and 7 after NP-OVA immunization (***Brookens et al., 2024***) and harvested at day 9. (**B**) Representative flow plots of GL7^+CD95^+GC B cells among viable IgD^neg B cells and

*Figure 7 continued on next page*

*Figure 7 continued*

(**C**) aggregated mean (± SEM) frequencies of such GC B cells from three replicate experiments (7 WT; 9 cKO mice). Mann-Whitney U test was used to calculate p values. (**D–F**) Effect of PexRAP on the levels of ROS (**D**), cell death (**E**), and proliferation (**F**) of GC B cells. (**D**) Total cellular ROS in IgD$^{neg}$ CD38$^{neg}$ GL7$^+$ GC B cells was determined by flow cytometry after staining with surface markers and H$_2$DCFDA as described in the Methods. The graph shows the mean (± SEM) geometric MFI of H$_2$DCFDA from two independent replicate experiments (n=6 WT and 6 cKO). (**E**) PexRAP promotes GC B cell survival. Shown are the representative flow plot (left), and a dot graph aggregating all experiments' outcomes for the frequencies of annexin V$^+$ 7AAD$^+$ cells in GC B gated cells as in Fig 7**D** (right panel). (**F**) PexRAP regulates proliferation of GC B cells. Tamoxifen-treated mice (*huCD20*-CreER$^{T2}$; *Dhrs7b$^{+/+}$* or *huCD20*-CreER$^{T2}$; *Dhrs7b$^{f/f}$*, i.e. *Dhrs7b Δ B*; shown as *Dhrs7bΔ/Δ−$^B$*) were immunized with SRBC, and the mice were injected with BrdU as described in Methods. Shown are representative histograms for WT and *Dhrs7bΔ/Δ* GC B cells as indicated (left panel) and a graph indicating the aggregated result of each independent experiment (right panel) (x=2; n=6 WT and 6 cKO).

approximate glycolytic activity - were unaffected (***Figure 9G***). Thus, the effects of PexRAP on B cell metabolism are not global, in that altered respiration did not reflect a decrease in glucose-stimulated acidification. Inasmuch as peroxisomes and mitochondria form contacts and functionally interact with the ER, we measured relative ER mass and found that the signal of the fluorophore ERTracker was consistently reduced in PexRAP-depleted B cells (***Figure 9H and I***). We infer that both mitochondrial function and ER mass in B cells are promoted by the product of the gene.

## Discussion

We have shown herein that B lymphocyte expression of an enzyme essential for biosynthesis of a subset of ether phospholipids is crucial for part of the increases in their levels localized to GC. Moreover, the genetic approach used to deplete B cells of PexRAP allowed us to show that the B cell type-restricted gene, *Dhrs7b*, promotes B cell survival, proliferation, and GC size along with affinity maturation and serum concentrations of Ag-specific Ab after immunization. Taken together, this work (i) establishes the predictive value of the discovery approach that identified an unexpected feature of cellular biochemistry in GC, (ii) indicates that B cell-autonomous generation of ether lipid species is crucial for B cell survival, and (iii) regulates the qualities of the Ab elicited by immunization.

Using imaging mass spectrometry for discovery-based hypothesis generation, we had reported that the concentrations of at least a dozen ether phospholipids are increased in splenic GC relative to the remainder of the tissue (***Jones et al., 2020***). The analyses presented here confirm and extend the observations and provide evidence that after immunization, these concentrations increase even in the primary B cell follicle, albeit to a modest degree when compared to the magnitude of increases in GC and in B cells activated ex vivo. The cell-type-specific gene inactivation after establishment of a pre-immune population establishes that an enzyme crucial for generation of a number of plasmalogens has substantial effects on B lymphocyte physiology and function. Plasmalogens and other ether lipids and phospholipids have long been shown to be major constituents of lipid bilayers and cell membranes (***Dean and Lodhi, 2018***; ***Honsho and Fujiki, 2023***; ***Braverman and Moser, 2012***). However, remarkably little is known about whether or not cell-intrinsic synthesis of ether lipids or the balance among their specific molecular species matters for hematopoietic cells or in immunity. Several inborn errors of metabolism in humans are attributable to loss-of-function mutations of genes encoding peroxisomal proteins that impact ether lipid (and hence plasmalogen) synthesis (***Gorgas et al., 2006***) but affect other critical processes as well [reviewed in ***Dean and Lodhi, 2018***; ***Gorgas et al., 2006***; ***Jiménez-Rojo and Riezman, 2019***; ***Rangholia et al., 2021***; ***Bozelli et al., 2021***; ***Wallner and Schmitz, 2011***]. Several of these - such as Zellweger Syndrome and rhizomelic chondrodysplasia punctata (RCDP) - and the mouse models generated to study mechanisms in these human diseases lead to severe neurological defects and early post-partum death (***Braverman and Moser, 2012***; ***Gorgas et al., 2006***). Whether or not lymphocyte development, homeostasis or function was affected in these studies is not clear. Deficient generation of ether lipids is among many abnormalities caused by elimination of fatty acid synthase (FAS) (***Lodhi et al., 2015b***; ***Lodhi et al., 2012***). Acute inactivation of *Fasn*, the gene encoding this enzyme, in young mature mice caused a decrease in spleen size disproportionate to the reduction in neutrophils, which was the most profound hematological consequence (***Lodhi et al., 2015b***). In addition, circulating lymphocytes were reduced despite normal steady-state representation in the marrow. In-trans cell-extrinsic functions of ether lipids have been noted previously - specifically, as ligands for the invariant natural killer T cell receptor (***Facciotti et al., 2012***) or for a G-protein-coupled receptor on natural killer cells (***Hossain et al., 2022***). As such, the

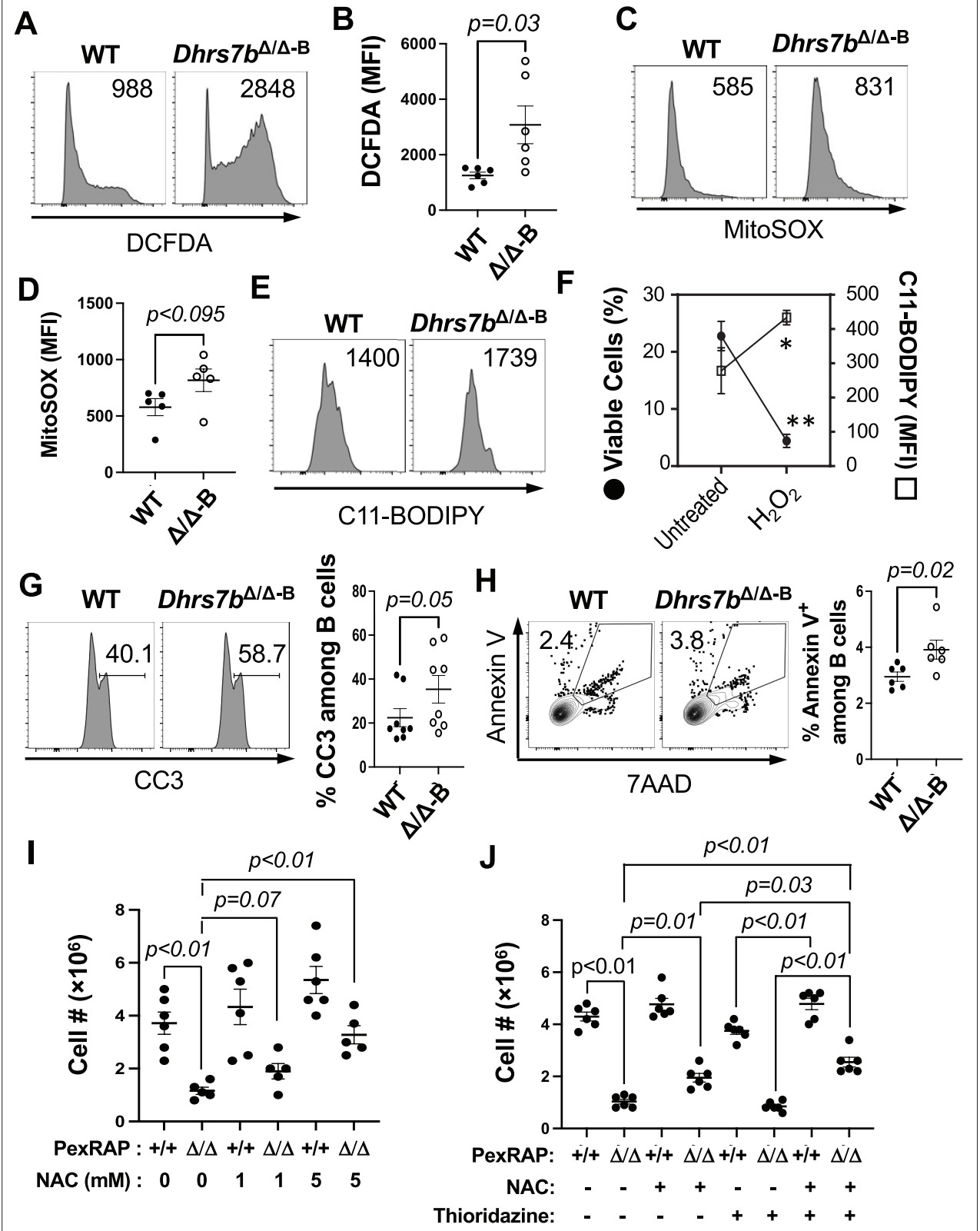

**Figure 8.** PexRAP contributes to ROS homeostasis and B cell population growth in vitro. (**A**–**D**) PexRAP is critical for maintenance of normal ROS levels. Bead-purified B cells (*Dhrs7bΔ/Δ* and *Dhrs7b+/+*) from spleens of tamoxifen-treated *huCD20*-CreER[T2] mice were cultured 3 days in anti-CD40, BAFF, IL-4, IL-5, and 4-hydroxytamoxifen. Total cellular (**A**, **B**) and mitochondrial ROS, mtROS (**C**, **D**) in B lymphoblasts were then determined by flow cytometry after staining with surface markers and H₂DCFDA and MitoSOX, as described in the Methods. Representative histogram image of H$_2$DCFDA (**A**) and

*Figure 8 continued on next page*

*Figure 8 continued*

MitoSOX (**C**) in the B cell gate, and aggregated mean (± SEM) geometric MFI of H$_2$DCFDA (**B**) and MitoSOX (**D**) from three independent experiments, each using two mice of each type (6 WT; 6 cKO). p values were calculated by Mann-Whitney U test. (**E**) PexRAP restrains lipid peroxidation. B cells were activated and cultured as in (**A**). A representative result of flow cytometric analyses of lipid peroxidation assayed using C11-Bodipy is shown, based on three independent experiments. (**F, G**) PexRAP promotes B cell survival. (**F**) Shown are the mean (± SEM) frequencies of total viable 'events' (by FSC, SSC, and 7-AAD exclusion) in flow cytometry (filled circles) and MFI of Bodipy-C11 (open squares) after exposure to H$_2$O$_2$ (200 µM). (*, ** - p=0.03 and 0.003, respectively). (**G–I**) WT and *Dhrs7bΔ/Δ* B cells were cultured (2 days) in anti-CD40, BAFF, IL-4, IL-5, and 4-hydroxytamoxifen. (**G**) Increased cleaved caspase 3 (CC3) in PexRAP-deficient cells generated in vitro. Cleaved caspase 3 in B cells was detected by intracellular staining and flow cytometry. Shown are a representative pair of histograms for WT and *Dhrs7bΔ/Δ* B cells as indicated (left panel) and a dot graph aggregating all experiments' outcomes (right panel). (**H**) Frequencies of apoptotic B cells in cultures as in (**A–D**) were scored for annexin V and 7AAD as described (*Lee et al., 2013*). Shown are representative data of flow plots in the lymphocyte gate (left panel) and a dot graph aggregating all experiments' outcomes (right panel). (**I, J**) PexRAP and ROS control promote B cell population growth in vitro. WT and *Dhrs7bΔ/Δ* B cells were cultured 5 days in anti-CD40, BAFF, IL-4, IL-5 and 4-hydroxytamoxifen in the presence or absence of ROS scavenger NAC (1 mM vs 5 mM) (**H**). (**I**) B cells activated as in (**A–G**) were cultured 5 days in the presence or absence of NAC (1 mM) or thioridazine (100 nM). The bar graphs show the mean (± SEM) recovered cell number from three independent experiments, each with two independent samples of each genotype (WT; *Dhrs7bΔ/Δ*). Complementary results of experiments including anti-IgM for BCR cross-linking are shown in *Figure 2—figure supplement 1*, *Figure 6—figure supplement 1*. Further related data are in *Figure 8—figure supplement 1*.

The online version of this article includes the following figure supplement(s) for figure 8:

**Figure supplement 1.** Distinct outcomes of heightened ROS elicited by H$_2$O$_2$ versus menadione.

widespread loss of FAS function and the potential for indirect and pleiotropic effects of this enzyme deficiency left unresolved what cells actually depend on their own synthesis of ether lipids.

PexRAP, the enzyme encoded by the *Dhrs7b* gene, generates intermediates vital for the synthesis of some - but not all - ether lipids in cells such as neutrophils (*Lodhi et al., 2015b*). We interpret our lipidomic data as indicating that this is the case for B lymphocytes. Specifically, although a number of plasmalogen or other ether lipid species in the naive, activated, or, in IMS, GC B cells were lower as a consequence of cell-type-specific gene inactivation, other phospholipids in these classes were unaffected. Analyses of neutrophil numbers and survival provided evidence that this enzyme can be crucial for a normal membrane composition. Moreover, PexRAP promoted the viability of this very short-lived cell type whose membranes have a high plasmalogen content (*Lodhi et al., 2015b*). The bloodborne population of lymphocytes was reduced in the *Rosa26*-CreER$^{T2}$, *Dhrs7b*$^{f/f}$ mice (*Lodhi et al., 2015b*), but the specific cell types were not reported and cells in circulation may not reflect the spleen or other organs. Of note, the interpretation that the neutrophil phenotype is due to insufficiency of ether lipids or plasmalogens has been questioned based on alternative gene disruption results (*Dorninger et al., 2015*). Several models of unconditional gene inactivation or mutation that drastically reduced the phospholipid end products were noted to have normal neutrophils summarized in *Dorninger et al., 2015*, albeit at older ages (4–7 months) than those used in *Lodhi et al., 2015b*. However, these caveats are subject to points articulated in *Lodhi et al., 2015a*. There may be age-related changes in the processes analyzed herein using young (age ~2 months) mice (*Lodhi et al., 2015a*). Moreover, cellular and organismal selection for adaptation variants makes the results of acute loss-of-function inherently different from unconditional loss and heterozygote parentage (*Lodhi et al., 2015a*). The potential for toxicity induced by 4OHT activation of CreER$^{T2}$ was among concerns mooted about the earlier study of neutrophils (*Dorninger et al., 2015*). This concern may have been extrapolated from papers that noted impairment in earlier hematopoiesis when *Rosa26*-CreER$^{T2}$ was combined with very intense regimens of tamoxifen administration to infant mice (*Higashi et al., 2009*; *Rossi et al., 2023*). Such studies used individual doses over fourfold higher than those used herein and delivered by gavage five times on a daily basis instead of three. Nonetheless, the IgM$^+$ B cell population sizes in spleen and marrow were normal even in the face of reduced erythropoiesis and thymopoiesis (*Higashi et al., 2009*). Moreover, the analyses herein compared B-lineage-specific deletion to CreER$^{T2+/-}$, *Dhrs7b*$^{+/+}$ mice identically dosed in parallel with the test subjects. Thus, our work indicates that PexRAP is essential for normal physiology and function of mature B lymphocytes, in particular after their activation.

The findings are most consistent with a capacity for PexRAP function to contribute towards B lymphocyte survival and effective proliferation by restraining both executioner caspase activation and lipid peroxidation due to excessive ROS levels. For instance, population expansion of *Dhrs7b Δ/Δ* B cells was severely undermined after mitogenic stimulation in vitro, with little difference in the division

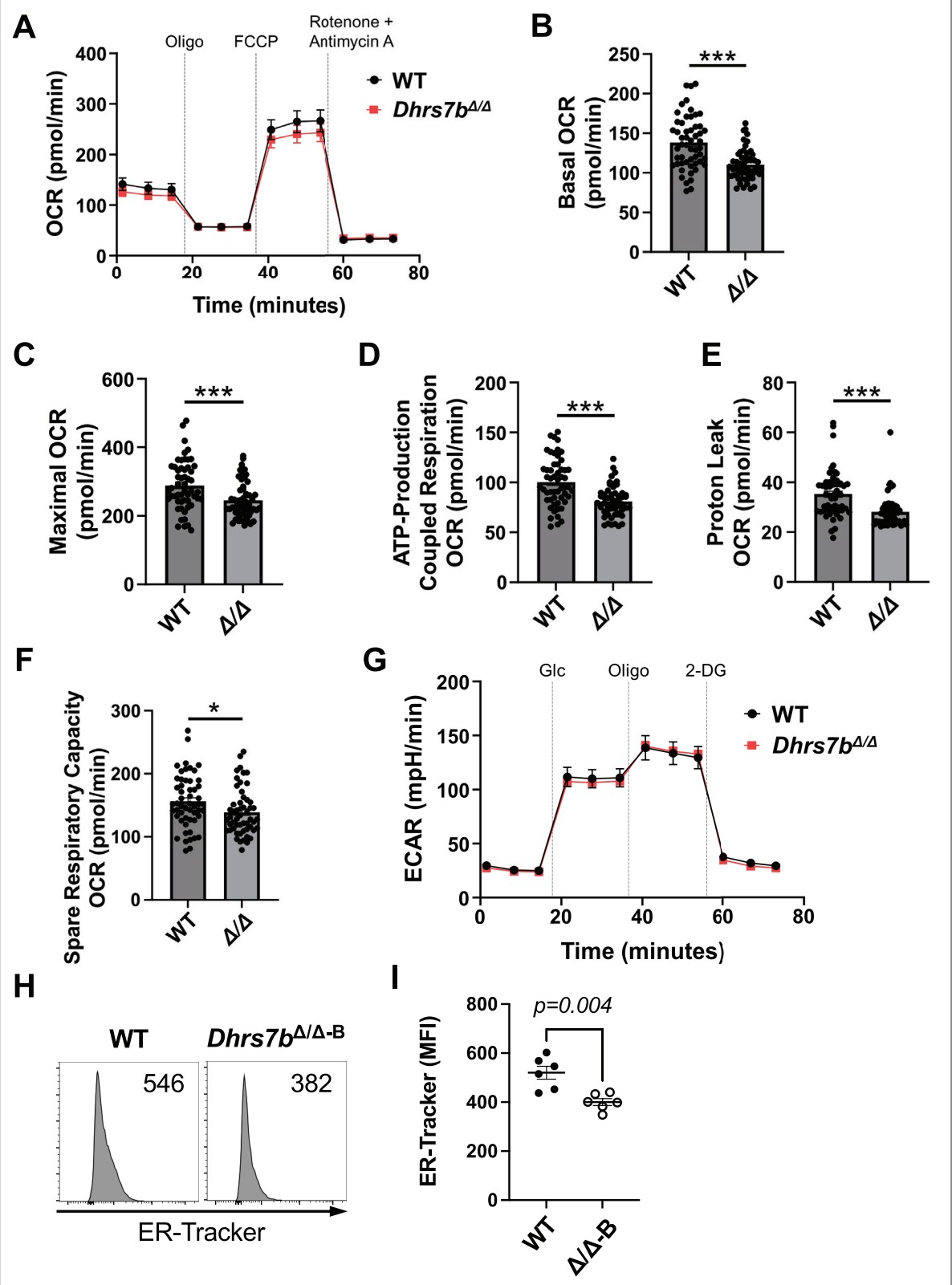

**Figure 9.** PexRAP deficiency in activated B cells reduces mitochondrial metabolism and ER mass. (A–G) Purified B cells were activated with and cultured (2 d) in anti-CD40, BAFF, IL-4, IL-5, and 4-OHT, then analyzed using a Seahorse XFe96 after harvest and division into equal portions. (A) Shown are aggregated results from three independent experiments measuring oxygen consumption rate (OCR) quantified via metabolic flux analysis during mitochondrial stress testing. (B) Basal OCR, (C) maximal OCR, (D) ATP-production coupled respiration, (E) proton leak, and (F) spare respiratory

*Figure 9 continued on next page*

*Figure 9 continued*

capacity of WT and *Dhrs7b*$^{\Delta/\Delta}$ B cells assayed in (**A**). Data points of individual samples are shown (each individual dot), as well as bars to display mean (± SEM) values. T-tests with Welch's correction were used to calculate p-values. \*, p<0.05; \*\*\*, p<0.001. (**G**) Extracellular acidification rate (ECAR) of B cells cultured as described previously (*Brookens et al., 2024*) and assessed using the glycolytic stress test. (**H**, **I**) PexRAP influences ER mass. WT and *Dhrs7b*$\Delta/\Delta$ B cells were cultured for 5 days in anti-CD40, BAFF, IL-4, IL-5, and 4-hydroxytamoxifen, followed by the staining with ER-Tracker Green. Representative histogram of ER-Tracker Green in viable cells (**H**) and aggregated MFI (± SEM) of ER-Tracker Green from three replicate experiments (**I**). Mann-Whitney U test was used to calculate p values.

history but higher frequencies of annexin V-positive B cells. These findings are consistent with those obtained with neutrophils (*Lodhi et al., 2015b*). On the surface, such results might appear to present a conundrum relative to work indicating that both fatty acyl-CoA reductase 1 and the enzyme that later generates an alkenyl bond at a terminal step in plasmalogen biosynthesis are crucial for the susceptibility of HT-1080 and 786-O tumor cell lines to ferroptosis (*Cui et al., 2021*). However, our data are consistent with published evidence that PexRAP is needed for a narrower subset of ether lipids than the broad requirement for FAS or fatty acyl-CoA reductases (*Lodhi et al., 2015b*; *Lodhi et al., 2012*; *Dean and Lodhi, 2018*; *Honsho and Fujiki, 2023*; *Lodhi and Semenkovich, 2014*; *He et al., 2021*). Moreover, molecular dissection of mechanisms in various non-hematopoietic cell types (including the cancer line 786-O) showed that ether lipid biosynthesis can either promote susceptibility or resistance to ferroptosis (*Zou et al., 2020*). Collectively, our findings suggest that for B cells, the inactivation of PexRAP tilts the balance toward death and reduced Ab - perhaps involving lower fractions of polyunsaturated fatty acids (PUFA; *Esselman et al., 2023*), or a lower ratio of plasmalogens (alkenyl linkages) relative to alkyl-linked ether lipids.

PexRAP localizes to peroxisomes (*Yazicioglu et al., 2023*; *Brookens et al., 2024*; *Olenchock et al., 2017*), and prior work has suggested both that peroxisomes are increased in GC B cells and likely contribute to FAO by these cells (*Chapman and Chi, 2022*). In line with the possibility that this organelle may affect PexRAP-dependent survival, our data indicated that specific inhibition of peroxisomal FAO could collaborate with sub-optimal ROS scavenging to mitigate the reduced growth of activated B cells. However, recent findings indicate that PexRAP can also be expressed in the ER and is capable of contributing to ether lipid biosynthesis via catalysis in this organelle in addition to or instead of the peroxisome (*Honsho et al., 2020*). Thus, although ROS generated in peroxisomes likely contribute to the stress experienced by PexRAP-deficient B cells, the phenotypes observed herein may not exclusively reflect a need for the peroxisomal fraction of the enzyme but instead might include functions of ER-localized protein.

Our findings with an intermediary in ether lipid biosynthesis add to an increasing body of evidence that metabolism in B lymphocytes affects their differentiation or function (*Jellusova, 2020*; *Boothby et al., 2022*; *O'Neill et al., 2016*; *Cho et al., 2016*; *Weisel et al., 2020*; *Chen et al., 2021*; *Urbanczyk et al., 2022*; *Yazicioglu et al., 2023*; *Brookens et al., 2024*). Many sources and forms of Ab can be protective, while in several autoimmune diseases, these molecules can drive pathology (*Geyer et al., 2021*; *McGettigan and Debes, 2021*). The capacity to generate increased affinities and circulating concentrations of Ab is an important factor both in protective immunity and autoimmune disease. Antigen-independent, TLR-driven PC differentiation requires increased oxidative phosphorylation (*Price et al., 2018*), while oxidative metabolism in GC B cells promotes affinity maturation by unknown mechanisms (*Chen et al., 2021*; *Urbanczyk et al., 2022*; *Yazicioglu et al., 2023*). While not the exclusive determinant of such properties, GCs are major sources of higher affinity Ab with a broader spectrum of specificities (*Boothby et al., 2019*; *Elsner and Shlomchik, 2020*; *Good-Jacobson and Shlomchik, 2010*). A limitation of our study is that for practical reasons, we were unable to use purified GC B cells for immunoblots or nucleic acid quantitation to determine the extent of counterselection or extent of *Dhrs7b* locus inactivation in these cells after their clonal expansion. Nonetheless, several lines of evidence suggest that PexRAP affects the physiology of GC B cells during their residence in this micro-anatomic structure. Ether lipids are considered to be short-lived, and the levels of several within the GC were reduced by PexRAP depletion prior to immunization. Moreover, rates of cell cycling as measured with BrdU incorporation appeared reduced, while ROS levels within GC B cells were higher along with the frequency of death. Because GC Tfh cells provide new stimulation to GC B cells that present peptide recognized by the TCR, some of these effects may be secondary consequences of a substantially lower GC-Tfh population even when B-cell-specific loss of function

was tested. A speculative corollary is that the observed reduction in ER mass might be functionally notable. For instance, cognate B-T interaction in the GC involves BCR-mediated internalization of Ag followed by processing and MHC-II presentation of T cell epitope peptide. Moreover, ether lipids and plasmalogens are presented to innate-like T cells via CD1d (*Facciotti et al., 2012*). In addition to the GC-localized abnormalities, GC formation, size, and function also depend on the efficiency of population growth for B cells in a phase before they enter and take on characteristics of GC B cells, and then homeostatic and differentiative processes while in the secondary follicle. Thus, while the findings with the post-immunization inactivation of *Dhrs7b* using *S1pr2*-CreER[T2] provide evidence that the expression of PexRAP is important within GC B cells, the decreases of GC and affinity-matured Ab are likely also to involve a pre-GC phase of clonal expansion. In any case, the rapidity of the effects suggests that pharmacological inhibition of PexRAP may be a target in inflammatory disease in which B cells participate, particularly in light of the potential for impacts on pro-inflammatory neutrophils.

# Materials and methods

## Key resources table

| Reagent type (species) or resource | Designation | Source or reference | Identifiers | Additional information |
|---|---|---|---|---|
| Genetic reagent (*Mus musculus*) | C56/Bl6(B6) - *Dhrs7b*[f/f] | Semenkovich (WUSTL); PMID:25565205 | none | conditional loss-of-function allele encoding PexRAP |
| Genetic reagent (*Mus musculus*) | B6 - *Rosa26*-CreER[T2] | Ludwig (Ohio State Univ.) PMID:16314438 | none | conditionally active recombinase |
| Genetic reagent (*Mus musculus*) | B6 - *huCD20*-CreER[T2] | Shlomchik (Yale Univ; now @ UPMC) PMID:22555432 | none | B-cell-specific conditionally active recombinase |
| Genetic reagent (*Mus musculus*) | B6 - *S1pr2*-CreER[T2] | Cyster (UCSF) PMID:27158841 | none | GCB (& GC-Tfh / neural)-expressed conditionally active recombinase |
| Peptide, recombinant protein | NP-ovalbumin | Biosearch Tech - now Fina Bio (*Thomas and Hulbert, 1996*; *De Silva and Klein, 2015*; *Cyster and Allen, 2019*; *Victora and Nussenzweig, 2022*; *Guttormsen et al., 1999*) | Catalog N5051-100 | |
| Peptide, recombinant protein | NP(~20)-BSA | Biosearch Tech - now Fina Bio | Catalog N5050H | |
| Peptide, recombinant protein | NP2-PSA | Biosearch Tech - now Fina Bio | Catalog N5050L | |
| Biological sample (Sheep) | Defibrinated Sheep Red Blood Corpuscles (SRBC) | Colorado Serum Co. | Catalog # 31122 | must use fresh (<2 weeks) |
| Antibody | Anti-mouse PexRAP Ab (rabbit polyclonal) | Proteintech | Catalog #15530–1-AP | WB (1:1000) |
| Antibody | Anti-β-actin Ab (mouse monoclonal) | Santacruz Biotech | Catalog # sc-517582 | WB (1:2000) |
| Antibody | Anti-mouse GL7-FITC (rat monoclonal) | BD Biosciences | Catalog # 562080; RRID:AB_10894953 | FACS (1:250) IHC (1:100) |
| Antibody | Anti-mouse IgM-FITC (rat monoclonal) | BD Biosciences | Catalog # 553437; RRID:AB_394857 | FACS (1:500) |
| Antibody | Anti-mouse GL7-PE (rat monoclonal) | BD Biosciences | Catalog # 561530; RRID:AB_10715834 | FACS (1:250) |
| Antibody | Anti-mouse CD5-FITC (rat monoclonal) | BD Biosciences | Catalog # 553020; RRID:AB_394558 | FACS (1:250) |

*Continued on next page*

*Continued*

| Reagent type (species) or resource | Designation | Source or reference | Identifiers | Additional information |
|---|---|---|---|---|
| Antibody | Anti-mouse CD86-PE (rat monoclonal) | BD Biosciences | Catalog # 561963; RRID:AB_10896971 | FACS (1: 250) |
| Antibody | Anti-mouse IgD-PE (rat monoclonal) | eBioscience | Catalog #12-5993-83 | FACS (1:500) IHC (1:200) |
| Antibody | Anti-mouse PD1-PE (rat monoclonal) | BD Biosciences | Catalog # 568260; RRID:AB_2916860 | FACS (1:500) |
| Antibody | Anti-mouse CD21-BV510 (rat monoclonal) | BioLegend | Catalog # 123437; RRID:AB_2876441 | FACS (1:500) |
| Antibody | Anti-mouse TCRβ-PE-Cy5 (hamster monoclonal) | Invitrogen | Catalog # 15-5961-82 | FACS (1:500) |
| Chemical compound | Ghost Dye-Violet 450 | Tonbo Bioscience | Catalog # 13–0868 T500 | FACS (1:500) |
| Chemical compound | Ghost Dye-Violet 510 | Tonbo Bioscience | Catalog # 13–0870 T100 | FACS (1:500) |
| Antibody | Anti-mouse CD11b-PerCP (rat monoclonal) | Tonbo Bioscience | Catalog # 65–0112 U025 | FACS (1:500) |
| Antibody | Anti-mouse CD11c-PerCP (hamster monoclonal) | Tonbo Bioscience | Catalog # 65–0114 U025 | FACS (1:500) |
| Antibody | Anti-mouse F4/80-PerCP (rat monoclonal) | Tonbo Bioscience | Catalog # 65–4801 U025 | FACS (1:500) |
| Antibody | Anti-mouse Gr1-PerCP (rat monoclonal) | Tonbo Bioscience | Catalog # 67–5931 U025 | FACS (1:500) |
| Antibody | Anti-mouse CD19-APC (rat monoclonal) | Tonbo Bioscience | Catalog # 20–0193 U025 | FACS (1:500) |
| Antibody | Anti-mouse CD19-PE-Cy7 (rat monoclonal) | Tonbo Bioscience | Catalog # 60–0193 U025 | FACS (1:500) |
| Antibody | Anti-mouse B220-APC-Cy7 (rat monoclonal) | Tonbo Bioscience | Catalog # 25–0452 U025 | FACS (1:500) |
| Antibody | Anti-mouse CD43-APC (rat monoclonal) | BD Biosciences | Catalog # 560663; RRID:AB_1727479 | FACS (1:500) |
| Antibody | Anti-mouse CXCR4-APC (rat monoclonal) | BD Biosciences | Catalog # 558644; RID:AB_1645219 | FACS (1:250) |
| Antibody | Anti-mouse CD38-eFluor 450 (rat monoclonal) | eBioscience | Catalog # 48-0388-42 | FACS (1:500) |
| Antibody | Anti-mouse CD95-Biotin (hamster monoclonal) | BD Bioscience | Catalog # 554256; RRID:AB_395328 | FACS (1:500) |

*Continued on next page*

*Continued*

| Reagent type (species) or resource | Designation | Source or reference | Identifiers | Additional information |
|---|---|---|---|---|
| Antibody | Anti-mouse CD35-Biotin (rat monoclonal) | BD Bioscience | Catalog # 553816; RRID:AB_395068 | IHC (1:200) |
| Antibody | Anti-mouse CD138-PE (rat monoclonal) | BD Bioscience | Catalog # 561070; RRID:AB_2033998 | FACS (1:500) |
| Antibody | Anti-mouse cleaved (active) caspase 3-FITC (rabbit monoclonal) | BD Bioscience | Catalog # 559341; RRID:AB_397234 | 10 µL/sample |
| Commercial Assay, Kit | BrdU-APC | BD Bioscience | Catalog # 552598; RRID:AB_2861367 | Cell proliferation kit |
| Commercial Assay, Kit | Bodipy 581/591 C11 | Invitrogen | Catalog # 552598 | Lipid peroxidation kit |
| Peptide, recombinant protein | Streptavidin-APC | BD Bioscience | Catalog # 554067; RRID:AB_10050396 | FACS (1:1000) IF (1:200) |
| Peptide, recombinant protein | Streptavidin-PE-Cy7 | BD Bioscience | Catalog # 557598 | FACS (1:1000) |
| Peptide, recombinant protein | Streptavidin-APC-Cy7 | BD Bioscience | Catalog # 554063; RRID:AB_10054651 | FACS (1:1000) |
| Peptide, recombinant protein | Annexin V-PE | BD Bioscience | Catalog # 560930 | FACS (1:100) |
| Chemical compound | 7-AAD | BD Bioscience | Catalog # 559925; RRID:AB_2869266 | Flow cytometry |
| Chemical compound | CellTrace Violet | Invitrogen | Catalog # C34557 | Flow cytometry |
| Chemical compound | H2DCFDA | Invitrogen | Catalog # D399 | Flow cytometry |
| Chemical compound | MitoSOX | Invitrogen | Catalog # M36008 | Flow cytometry |
| Chemical compound | ER-Tracker Red | Invitrogen | Catalog # E34250 | Flow cytometry |
| Chemical compound | N-acetyl L-cysteine | Sigma Aldrich | Catalog # 616-91-1 | Cell culture |
| Chemical compound | Thioridazine Hydrochloride | Sigma Aldrich | Catalog # 130-61-0 | Cell culture |
| Chemical compound | Menadione | Sigma Aldrich | Catalog # 158-27-5 | Cell culture |
| Commercial Assay, Kit | Seahorse XF Glycolysis stress test | Agilent | Catalog # 103020–100 | Metabolic flux analysis kit |
| Commercial Assay, Kit | Seahorse XF Mito stress test | Agilent | Catalog # 103016–100 | Metabolic flux analysis kit |
| Software | FlowJo | TreeStar ->BD Biosciences | | Flow cytometry data analysis |

## Reagents

Monoclonal antibodies (mAb) against mouse CD40, CD90.2, B220 (CD45R), and other mAbs (purified, biotinylated, or fluorophore-conjugated) were from BD Biosciences or Tonbo Biosciences (San Diego CA) unless otherwise indicated. BAFF was from AdipoGen (San Diego, CA). Recombinant mouse IL-4 and recombinant mouse IL-5 were from Peprotech (Rocky Hill, NJ). Glucose, 2-deoxyglucose, oligomycin, rotenone, antimycin A, carbonyl cyanide 4-phenylhydrazone (FCCP), N-acetyl-cysteine (NAC), thioridazine hydrochloride, $H_2O_2$, and 4-hydroxytamoxifen (4-OHT) were from Sigma-Aldrich Chemicals (St. Louis, MO), and menadione from Cayman Chemicals (Ann Arbor, MI). Tamoxifen (Tmx) was from APExBio Technology (Houston TX). NP-BSA and NP-PSA (for capture in ELISA) as well as NP-OVA (for immunization) were obtained from Biosearch Technology (Novato, CA). SRBCs (sheep red blood cells) were from Thermo Fisher Scientific (Waltham, MA). CellTrace Violet cell proliferation kit, CM-H2DCFDA, MitoSOX Red, and BODIPY 581/591 C11 were obtained from Invitrogen (Waltham, MA).

## Mice, immunizations, ELISA, and ELISpot

All animal protocols were reviewed and approved by the Vanderbilt University Institutional Animal Care and Use Committee. Mice were housed in ventilated micro-isolators under Specified Pathogen-Free conditions in a Vanderbilt University Medical Center mouse facility and used both male and female mice at 6–8 weeks of age. For cell-type-specific inactivation of the *Dhrs7b* gene, *Dhrs7b*[f/f] mice (*Lodhi et al., 2015b*) were crossed with *Rosa26*-CreER[T2] (*Lodhi et al., 2015b*), huCD20-CerER[T2] (*O'Neill et al., 2016*), or *S1pr2*-CreER[T2] strains (*Dorninger et al., 2022*; *Kuerschner et al., 2012*), all on C57BL/6 genetic background. Tamoxifen was administered as previously reported (*Mayer et al., 2017*; *O'Neill et al., 2016*). To control for potential Cre toxicity, CreER[T2] mice were similarly injected to use as wild-type (WT) controls. Mice (ages ~8 weeks) were immunized with SRBCs ($2\times10^8$ cells per mouse) and analyzed 1 week after immunization as described (*Mayer et al., 2017*; *Jones et al., 2020*). Alternatively, mice were immunized and boosted by i.p. injections (each of 100 µg NP$_{16}$-OVA [Biosearch Technologies, Novato, CA] emulsified in 100 µL of Imject Alum [Thermo Fisher Scientific, Pittsburgh, PA]) as described previously (*Mayer et al., 2017*; *O'Neill et al., 2016*), and harvested for analyses 1 week after the second injection. Isotype-specific relative levels of Ag-specific Ab were quantitated by capture ELISA using Ag (NP$_{20}$-BSA or NP$_2$-PSA for all- or high-affinity hapten-specific Abs, respectively) followed by SBA Clonotyping System (Southern Biotech, Birmingham AL), as described previously (*Mayer et al., 2017*; *O'Neill et al., 2016*). As previously described (*O'Neill et al., 2016*), frequencies of Ab-secreting cells (ASCs) were analyzed by ELISpot and quantitated using an Immuno-Spot Analyzer (Cellular Technology, Shaker Heights, OH).

## Cell cultures, proliferation assay, and reversion of population growth

Splenic B cells were purified (90–95%) using negative selection as previously described (*O'Neill et al., 2016*) or by positive selection with anti-mouse B220 microbeads (Miltenyi Biotech, Auburn, CA). B cells ($5\times10^5$ cells in 1 mL) were activated with anti-CD40 (1 µg/mL) and cultured with BAFF (10 ng/mL), IL-4 (10 ng/mL), IL-5 (10 ng/mL), and 4-OHT (50 nM) in Iscove's Modified Dulbecco's Medium (IMDM) supplemented with 10% Fetal Bovine Serum (FBS), 100 U/mL penicillin, 100 µg/mL streptomycin (Invitrogen), 3 mM L-glutamine, nonessential amino acids (Invitrogen), 0.1 mM 2-ME (Sigma-Aldrich). To analyze the effect of ROS scavenger and inhibition of peroxisomal lipid oxidation on population growth, bead-purified B cells from tamoxifen-treated huCD20-CerER[T2] or *Dhrs7b*[f/f]; huCD20-CerER[T2] mice were activated with anti-CD40, cultured with BAFF, IL-4 and IL-5 in the presence or absence of NAC (1 mM and 5 mM) and/or thioridazine·HCl (100 µM) for 5 days. For enhancement of ROS in WT B cells, purified B cells were analyzed 3 days after activation as for ROS scavenging, but vehicle, $H_2O_2$ (200 µM) or menadione (8 µM) was added at three different time points (overnight, 3 hr, and 1 hr) prior to processing for flow cytometry to counting and measurements of DCFDA, annexin V, and C11 Bodipy signals.

For the analysis of proliferation in vitro, B cells were purified by depleting CD90.2$^+$ T cells and CD138$^+$ cells using biotinylated anti-CD90.2 Ab and biotinylated anti-CD138 Ab followed by streptavidin-conjugated microbeads (iMag; BD Bioscience), labeled with 5 µM CellTrace Violet (CTV, Invitrogen), activated and cultured as above. To analyze proliferation in vivo, B cells were purified, labeled with CTV, and injected intravenously into B-cell-deficient µMT recipient mice. Mice were harvested at 4 days after adoptive transfer. To measure proliferation rates of GC B cells, in vivo BrdU incorporation was performed by injecting mice with BrdU (2 mg/mouse; BD-Pharmingen, San Jose, CA) at 16 hr and 4 hr before harvesting spleens from SRBC-immunized mice. Cells were stained for surface markers (B220, IgD, GL7, CD38) followed by fixation, permeabilization, and staining with anti-BrdU-APC Ab using the BrdU Flow Kit (BD-Pharmingen) according to manufacturer's protocol.

## Flow cytometry - general and measurements of ROS, lipids peroxidation, and ER

GC B cells were identified as GL7$^+$ CD95$^+$ events in the viable B220$^+$ dump$^-$ gate (dump channel consisting of one fluorophore for CD11b, CD11c, F4/80, Gr1, TCRβ, and 7AAD), while plasmablasts were defined as B220$^+$ CD138$^+$ TACI$^+$ dump$^-$ cells. For flow analyses of total intracellular ROS and mitochondrial superoxide, B cells ($1\times10^6$ cells) were washed with PBS and stained with 1.25 µM CM-H$_2$DCFDA or 5 µM MitoSOX Red in PBS (20 min at 37 °C), respectively, then washed with 1% BSA containing PBS, and further stained with anti-B220, anti-CD19, anti-CD138, and Ghost-BV510.

To measure the lipid peroxidation, cultured B cells ($1\times10^6$) were washed with PBS and stained with 1.25 µM C11-BODIPY in PBS (20 min at 37 °C), and washed with 1% BSA containing PBS followed by surface staining as above. To analyze the level of endoplasmic reticulum (ER), B cells ($1\times10^6$) were stained with 0.5 µM ER-Tracker Green (Molecular Probes, Eugene OR) in Hank's Balanced Salt Solution with calcium and magnesium (HBSS/Ca/Mg; Gibco) for 30 min at 37 °C.

For measurements of cell death (*Hossain et al., 2022*), cultured B cells ($1\times10^6$) were washed twice with PBS, once with Annexin V binding buffer (10 mM HEPES, pH 7.4, 140 mM NaCl, 2.5 mM $CaCl_2$), and then incubated (15 min at 20 C in the dark) with Annexin V-rPE, 7AAD, APC-conjugated CD138, APC-Cy7-conjugated B220, e450-CD19. Cells were then washed with Annexin V binding buffer and analyzed on the flow cytometer. To measure levels of cleaved caspase-3 (*Mayer et al., 2017*), an activated apoptosis executor, the cultured cells were rinsed with PBS after harvest, stained (15 min at 4 C) with 7AAD, APC-conjugated CD138, APC-Cy7-conjugated B220, e450-CD19, washed (FBS, 1% v/v in PBS), then fixed with paraformaldehyde (4% w/v, 10 min at 20 C), permeabilized with permeabilization buffer (saponin, 0.2% w/v with FBS 1% in PBS), stained with FITC-conjugated Ab specific for cleaved caspase-3 (BD Biosciences), rinsed, and analyzed by flow cytometry.

## Immunohistochemistry, MALDI-IMS, and IMS data analysis

Mice were immunized with SRBC, and spleens were harvested at 1 week after immunization. Spleens were snap-frozen on dry ice and mounted to a cryostat chuck with a minimal amount of OCT (Thermo Fisher Scientific, San Jose, CA). Frozen spleens were cut at 12 µm thickness using a Leica CM3050S (Leica, IL, USA) at –20 °C, fixed with 1% paraformaldehyde for 10 min, washed twice with PBS, and blocked with M.O.M (Vector Lab) followed by incubation with GL7-FITC, anti-IgD-PE, and anti-CD35-biotin Ab followed by streptavidin-conjugated Alex647 at 4 °C. After the tile scanning of spleen sections, GC size and numbers were quantified with FIJI Image J software.

For Matrix-Assisted Laser Desorption/Ionization (MALDI) Imaging Mass Spectrometry (IMS), frozen spleens were cut as above and mounted onto indium tin oxide (ITO) coated microscope slides (Delta Technologies ETC). Slides were then vacuum desiccated for at least 30 min before matrix application. Pre-IMS autofluorescence (AF) images were acquired prior to matrix application on a Zeiss Axio Scan Z1 Slide Scanner using the brightfield channel and the DAPI (ex. 340–380 nm; em. 435–485 nm, blue), EGFP (ex. 450–490 nm; em. 500–550 nm, green), DsRed (ex. 538–562 nm; em. 570–640 nm, red) filter cubes. After taking AF images, a custom in-house developed sublimation device was used to apply the matrices 2,5-dihydroxyacetophenone (DHA) and 1,5-diaminonaphthalene (DAN) (Sigma Aldrich, St. Louis, MO, USA) to tissue sections for positive and negative ion mode analyses, respectively (*Hossain et al., 2022*). For immunized AID-GFP mice, the fluorescence emissions identified GC localization within the section used next for IMS. Alternatively, two serial sections of immunized mice spleens were used, one for 2D-IMS and the contiguous section for immunohistochemistry (IHC). Fluorescence data were registered with the 2D-IMS visualizations of phospholipids to align images (*Jones et al., 2020*), allowing quantitation of the ion intensities in specific m/z peaks within defined micro-anatomic portions of the spleen (*Figure 4B*). MALDI IMS data were acquired with a 10 µm pixel size (laser spot size 8 µm) in full scan mode using a Bruker trapped ion mobility time-of-flight (timsTOF) Flex mass spectrometer (Bruker Daltonics Billerica, MA, USA). Data were acquired with 250 shots per pixel and a mass range of *m/z* 200–2000. SCiLS (software Bruker Daltonics) was used to process the data, normalize ion intensity, visualize ion images, and merge the images.

## LC-MS for lipidomics

To prepare samples, a one-phase method was used to extract lipids (*Pellegrino et al., 2014*). Briefly, 0.5 mL of MeOH/MTBE/$CHCl_3$ mix (1.3:1:1) was added to a frozen pellet of B-cells ($1\times10^6$ cells total), spiked with 10 µL EquiSPLASH-lipidomics internal standard mix (Avanti Research), briefly vortexed and shaken gently for 20 min, followed by centrifugation at 20,000 x *g* for 15 min at 10 °C. The supernatant was transferred to a clean Eppendorf tube, evaporated under a gentle stream of $N_2$ gas, and resuspended in 100 µL methanol/$CHCl_3$ (9:1), and 2 µL were used for LC-HRMS (high-resolution MS) analysis. Each sample was injected two times - one injection in positive ESI mode followed by one in negative mode. Pooled QCs were injected to assess the performance of the LC and MS instruments at the beginning, in the middle and at the end of each sequence. Discovery lipidomics data were

acquired using a Vanquish UHPLC (ultrahigh performance liquid chromatography) system interfaced to a Q Exactive HF quadrupole/orbitrap mass spectrometer (Thermo Fisher Scientific).

Chromatographic separation was performed with a reverse-phase Acquity BEH C18 column (1.7 mm, 2.1x150 mm, Waters, Milford, MA) at a flow rate of 250 µL/min. Mobile phases were made up of 10 mM ammonium formate and 0.1% formic acid in (A) $H_2O/CH_3CN$ (40:60) and in (B) $CH_3CN/$ iPrOH (10:90). Gradient conditions were as follows: 0–1 min, B=20%; 1–8 min, B=20–100%; 8–10 min, B=100%; 10–10.5 min, B=100–20%; 10.5–15 min, B=20%. The total chromatographic run time was 15 min. Mass spectra were acquired over a precursor ion scan range of m/z 200–1600 at a resolving power of 60,000 using the following HESI-II source parameters: spray voltage 4 kV (3 kV in negative mode); capillary temperature 250 °C; S-lens RF level 60 V; $N_2$ sheath gas 40; $N_2$ auxiliary gas 10; auxiliary gas temperature 350 °C. MS/MS spectra were acquired for the top-seven most abundant precursor ions with an MS/MS AGC target of 1e5, a maximum MS/MS injection time of 100ms, and a normalized collision energy of 15, 30, 40. High-resolution mass spectrometry data were processed with MS-DIAL version 4.70 in lipidomics mode (*Tsugawa et al., 2020*). MS1 and MS2 tolerances were set to 0.01 and 0.025 Da, respectively. Minimum peak height was set to 30,000 to decrease the number of false positive hits. Peaks were aligned on a quality control (QC) reference file with RT tolerance of 0.1 min and mass tolerance of 0.015 Da. Default lipid library was used (Msp20210527163602_ converted.lbm2), solvent type was set to $HCOONH_4$ to match the solvent used for separation, and the identification score cut-off was set to 80%. All lipid classes were made available for the search. After lipid identification was completed, MS-DIAL results were exported into Excel and cleaned using minimum RSD for QC samples set to 20% and minimum ratio of QC to Blank set to 10. For species identified in both PexRAP-depleted (*Dhrs7b* Δ/Δ) B cells and controls, mean levels (areas under peak curves) and their variance were analyzed in Excel.

## Metabolic flux analyses

Oxygen Consumption Rate (OCR) was measured using Seahorse XF96 extracellular flux analyzer (Agilent Technology, Santa Clara, CA) as described previously (*O'Neill et al., 2016*). Briefly, in vitro activated B cells were washed twice, resuspended in XF Base Media (Agilent Technologies) supplemented with 2 mM L-glutamine, and equal numbers of B cells ($2 \times 10^5$) were plated on extracellular flux assay plates (Agilent Technologies) coated with 2.5 µg/mL CellTak (Corning) according to the manufacturer's protocol. Before extracellular flux analysis, B cells were rested (25 min at 37 °C, atmospheric $CO_2$) in XF Base Media. OCR and ECAR were measured before and after the sequential addition of 1.5 µM oligomycin, 0.5 µM FCCP and 0.5 µM rotenone/antimycin A.

## Acknowledgements

The experimental work was funded by NIH Grants to VUMC (R01 AI113292 [MRB], R01 HL106812 [MRB], followed by R21 AI164760 and R01 AI149722 [MRB] and Vanderbilt University Medical Center Pathology-Microbiology-Immunology departmental funds). Mass spectrometry imaging was supported in part by NIH grant P41 GM103391 to Vanderbilt University (RMC). Additional support for MA J was provided by the National Science Foundation, NSF DGE-1445197. NIH Shared Instrumentation Grant 1 S10 OD018015 as well as scholarships via the Cancer Center Support Grant (CA068485) and Diabetes Research Center (DK0205930) helped defray costs of Vanderbilt Cores. We thank L Clark for a critical reading and comments on the revised manuscript, J Cyster for generously expediting shipment of S1pr2 CreERT2 breeding stock, and Vanderbilt institutional cores (High-Throughput Screening; Flow Cytometry Shared Resource; Mass Spectrometry Shared Resource; Cell & Developmental Biology) for equipment, expertise, and assistance.

# Additional information

## Funding

| Funder | Grant reference number | Author |
| --- | --- | --- |
| National Institute of Allergy and Infectious Diseases | R21 AI164760 | Sung Hoon Cho<br>Kaylor Meyer<br>David M Anderson<br>Sergiy Chetyrkin<br>M Wade Calcutt<br>Mark R Boothby |
| National Institute of Allergy and Infectious Diseases | R01 AI149722 | Sung Hoon Cho<br>Kaylor Meyer<br>Sergiy Chetyrkin<br>M Wade Calcutt<br>Mark R Boothby |
| National Institute of General Medical Sciences | P41 GM103391 | Marissa A Jones<br>David M Anderson<br>Richard M Caprioli |
| National Science Foundation | Graduate Research Fellowship Program DGE-1445197 | Marissa A Jones |
| National Institutes of Health | R01 AI113292 | Sung Hoon Cho<br>Mark R Boothby |
| National Institutes of Health | R01 HL106812 | Sung Hoon Cho<br>Mark R Boothby |
| National Institutes of Health | 1 S10 OD018015 | Kaylor Meyer<br>Sung Hoon Cho<br>Mark R Boothby |
| National Institutes of Health | CA068485 | Kaylor Meyer<br>Sung Hoon Cho<br>Mark R Boothby |
| Diabetes Research Center | DK0205930 | Kaylor Meyer<br>Sung Hoon Cho<br>Mark R Boothby |

The funders had no role in study design, data collection and interpretation, or the decision to submit the work for publication.

## Author contributions

Sung Hoon Cho, Conceptualization, Data curation, Formal analysis, Supervision, Validation, Investigation, Visualization, Methodology, Writing – original draft, Writing – review and editing; Marissa A Jones, Conceptualization, Data curation, Formal analysis, Validation, Investigation, Visualization, Methodology; Kaylor Meyer, Data curation, Formal analysis, Investigation, Visualization, Methodology, carefully went through text, especially Methods and Legends, pertaining to the experiments he conducted; David M Anderson, Conceptualization, Data curation, Formal analysis, Investigation, Visualization, Methodology; Sergiy Chetyrkin, Data curation, Formal analysis, Investigation, Methodology; M Wade Calcutt, Formal analysis, Supervision, Investigation, Methodology, supervised Dr. Chetyrkin in the generation and first-level organizationi of lipidomics with purified B cells using LC-MS-MS analyses; Richard M Caprioli, Supervision, Funding acquisition, Doctoral thesis advisor of Marissa Jones, who conducted the work and had supervisory advice and guidance from Dr. Caprioli in performing the experiments of Figure 1 and Fig 4B; Clay F Semenkovich, Investigation, Methodology, Writing – review and editing, Prof. Semenkovich, in addition to providing both breeding stock and experiment-ready Rosa26-CreERT2, Dhrs7b f/f mice, provided guidance on tamoxifen use in this line, hepatic and other aspects of the system, and editorial input into the writing of the manuscript; Mark R Boothby, Conceptualization, Formal analysis, Supervision, Funding acquisition, Investigation, Visualization, Methodology, Writing – original draft, Project administration, Writing – review and editing

## Author ORCIDs
Sung Hoon Cho  https://orcid.org/0000-0002-6463-4032
Clay F Semenkovich  https://orcid.org/0000-0003-1163-1871
Mark R Boothby  https://orcid.org/0000-0003-2593-8276

## Ethics

This study was performed in accordance with the recommendations in the Guide for the Care and Use of Laboratory Animals of the National Institutes of Health. All of the animals were handled according to approved institutional animal care and use committee (IACUC) protocols of the Vanderbilts (Vanderbilt University, Inc; Vanderbilt University Medical Center, Inc). The specific approved protocols included authorization for all procedures described herein, viz. genotyping, injections for chemical activation of modified Cre recombinase and for immunizations with the immunogens reported here, as well as for phlebotomy and euthanasia in line with current best practices (NIH OLAW).

Reviewer #1 (Public review): https://doi.org/10.7554/eLife.104580.3.sa1
Reviewer #2 (Public review): https://doi.org/10.7554/eLife.104580.3.sa2
Author response https://doi.org/10.7554/eLife.104580.3.sa3

---

# Additional files

## Supplementary files
MDAR checklist

Source data 1. Tabulations of root data used to prepare graphs for figure panels, or images (e.g., of immunoblots) for that purpose.

## Data availability

This work generated lipidomic and imaging mass spectrometric data. Lipidomic (LC-MS-MS) data with B cells are deposited at Metabolights as project MTBLS13880, accessed via the search tool at https://www.ebi.ac.uk/metabolights/ (*Yurekten et al., 2024*). The dataset is freely accessible. The IMS data also are freely available. A series of six compressed file folders ranging in downloaded size from ~31 GB to ~97 GB and totaling ~405 GB are available without password protection or restriction to access and downloads via https://www.dropbox.com/scl/fo/37hxp7wvgav815se ozugk/AJoQMnyyErmoFi2cDncsYLA?rlkey=7yvle0w7jroj45rq72onljuii&e=1&st=wg6mf772&dl=0, and the Vanderbilt University Mass Spectrometry Research Center is committed to continuation of this freely accessible Dropbox. The nature of the data is such that most cannot be sub-divided into parcels that are within the size limits that can be handled by the eLife-recommended data repositories. The subset of the data that can be uploaded into eLife-recommended repositories (due to limits on file size and project size) is publicly available at the Open Science Framework (https://doi.org/10.17605/OSF.IO/GCP3M). With one additional entry (~32 GB), data also are available in Zenodo (https://doi.org/10.5281/zenodo.19672730). In addition to the commitment of the Mass Spectrometry Research Center of Vanderbilt University to long-term maintenance of the publicly accessible Dropbox entry, all such data are also preserved by the Mass Spectrometry Research Center of Vanderbilt University using backed-up servers. If unable to access the data via the repositories noted above (including Dropbox), stored data are freely available upon request (david.m. anderson@vanderbilt.edu).

The following datasets were generated:

| Author(s) | Year | Dataset title | Dataset URL | Database and Identifier |
|---|---|---|---|---|
| Boothby MR, Fourkin SCV, Chetyrkin S | 2026 | B cell expression of the enzyme PexRAP, an intermediary in ether lipid biosynthesis, promotes antibody responses and germinal center size. | https://www.ebi.ac.uk/metabolights/editor/MTBLS13880/overview | Metabolights, MTBLS13880 |
| Boothby MR | 2026 | B cell expression of an enzymatic intermediary in ether lipid biosynthesis promotes antibody responses and germinal center size | https://doi.org/10.17605/OSF.IO/GCP3M | Open Science Framework, 10.17605/OSF.IO/GCP3M |
| Boothby MR, Anderson DM, Cho SH | 2026 | Imaging Mass Spectrometry _primary data_spatial lipidomic analysis of mouse spleen | https://doi.org/10.5281/zenodo.19672730 | Zenodo, 10.5281/zenodo.19672730 |

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
