## [Editor Report · eLife Assessment]

This study provides **useful** insights into the ways in which germinal center B cell metabolism, particularly lipid metabolism, affects cellular responses. The authors use sophisticated mouse models to **convincingly** demonstrate that ether lipids are relevant for B cell homeostasis and efficient humoral responses. The authors then conducted in vivo as well as in vitro experiments, thereby strengthening their conclusions.

---

## [Referee Report · Reviewer #1 (Public review)]

In this manuscript, Hoon Cho et al. present a novel investigation into the role of PexRAP, an intermediary in ether lipid biosynthesis, in B cell function, particularly during the Germinal Center (GC) reaction. The authors profile lipid composition in activated B cells both in vitro and in vivo, revealing the significance of PexRAP. Using a combination of animal models and imaging mass spectrometry, they demonstrate that PexRAP is specifically required in B cells. They further establish that its activity is critical upon antigen encounter, shaping B cell survival during the GC reaction.

Mechanistically, they show that ether lipid synthesis is necessary to modulate reactive oxygen species (ROS) levels and prevent membrane peroxidation.

Highlights of the Manuscript:

The authors perform exhaustive imaging mass spectrometry (IMS) analyses of B cells, including GC B cells, to explore ether lipid metabolism during the humoral response. This approach is particularly noteworthy given the challenge of limited cell availability in GC reactions, which often hampers metabolomic studies. IMS proves to be a valuable tool in overcoming this limitation, allowing detailed exploration of GC metabolism.

The data presented is highly relevant, especially in light of recent studies suggesting a pivotal role for lipid metabolism in GC B cells. While these studies primarily focus on mitochondrial function, this manuscript uniquely investigates peroxisomes, which are linked to mitochondria and contribute to fatty acid oxidation (FAO). By extending the study of lipid metabolism beyond mitochondria to include peroxisomes, the authors add a critical dimension to our understanding of B cell biology.

Additionally, the metabolic plasticity of B cells poses challenges for studying metabolism, as genetic deletions from the beginning of B cell development often result in compensatory adaptations. To address this, the authors employ an acute loss-of-function approach using two conditional, cell-type-specific gene inactivation mouse models: one targeting B cells after the establishment of a pre-immune B cell population (Dhrs7b^f/f, huCD20-CreERT2) and the other during the GC reaction (Dhrs7b^f/f; S1pr2-CreERT2). This strategy is elegant and well-suited to studying the role of metabolism in B cell activation.

Overall, this manuscript is a significant contribution to the field, providing robust evidence for the fundamental role of lipid metabolism during the GC reaction and unveiling a novel function for peroxisomes in B cells.

Comments on revisions:

There are still some discrepancies in gating strategies. In Fig. 7B legend (lines 1082-1083), they show representative flow plots of GL7+ CD95+ GC B cells among viable B cells, so it is not clear if they are IgDneg, as the rest of the GC B cells aforementioned in the text.

Western blot confirmation: We understand the limitations the authors enumerate. Perhaps an RT-qPCR analysis of the Dhrs7b gene in sorted GC B cells from the S1PR2-CreERT2 model could be feasible, as it requires a smaller number of cells. In any case, we agree with the authors that the results obtained using the huCD20-CreERT2 model are consistent with those from the S1PR2-CreERT2 model, which adds credibility to the findings and supports the conclusion that GC B cells in the S1PR2-CreERT2 model are indeed deficient in PexRAP

Lines 222-226: We believe the correct figure is 4B, whereas the text refers to 4C.

Supplementary Figure 1 (line 1147): The figure title suggests that the data on T-cell numbers are from mice in a steady state. However, the legend indicates that the mice were immunized, which means the data are not from steady-state conditions.

---

## [Referee Report · Reviewer #2 (Public review)]

Summary:

In this study, Cho et al. investigate the role of ether lipid biosynthesis in B cell biology, particularly focusing on GC B cell, by inducible deletion of PexRAP, an enzyme responsible for the synthesis of ether lipids.

Strengths:

Overall, the data are well-presented, the paper is well-written and provides valuable mechanistic insights into the importance of PexRAP enzyme in GC B cell proliferation.

Weaknesses:

More detailed mechanisms of the impaired GC B cell proliferation by PexRAP deficiency remain to be further investigated. In minor part, there are issues for the interpretation of the data which might cause confusions by readers.

Comments on revisions:

The authors improved the manuscript appropriately according to my comments.

---

## [Author Response]

The following is the authors’ response to the current reviews.

**Reviewer #1 (Public review):**
In this manuscript, Hoon Cho et al. present a novel investigation into the role of PexRAP, an intermediary in ether lipid biosynthesis, in B cell function, particularly during the Germinal Center (GC) reaction. The authors profile lipid composition in activated B cells both in vitro and in vivo, revealing the significance of PexRAP. Using a combination of animal models and imaging mass spectrometry, they demonstrate that PexRAP is specifically required in B cells. They further establish that its activity is critical upon antigen encounter, shaping B cell survival during the GC reaction. Mechanistically, they show that ether lipid synthesis is necessary to modulate reactive oxygen species (ROS) levels and prevent membrane peroxidation.Highlights of the Manuscript:The authors perform exhaustive imaging mass spectrometry (IMS) analyses of B cells, including GC B cells, to explore ether lipid metabolism during the humoral response. This approach is particularly noteworthy given the challenge of limited cell availability in GC reactions, which often hampers metabolomic studies. IMS proves to be a valuable tool in overcoming this limitation, allowing detailed exploration of GC metabolism.The data presented is highly relevant, especially in light of recent studies suggesting a pivotal role for lipid metabolism in GC B cells. While these studies primarily focus on mitochondrial function, this manuscript uniquely investigates peroxisomes, which are linked to mitochondria and contribute to fatty acid oxidation (FAO). By extending the study of lipid metabolism beyond mitochondria to include peroxisomes, the authors add a critical dimension to our understanding of B cell biology.Additionally, the metabolic plasticity of B cells poses challenges for studying metabolism, as genetic deletions from the beginning of B cell development often result in compensatory adaptations. To address this, the authors employ an acute loss-of-function approach using two conditional, cell-type-specific gene inactivation mouse models: one targeting B cells after the establishment of a pre-immune B cell population (Dhrs7b^f/f, huCD20-CreERT2) and the other during the GC reaction (Dhrs7b^f/f; S1pr2-CreERT2). This strategy is elegant and well-suited to studying the role of metabolism in B cell activation.Overall, this manuscript is a significant contribution to the field, providing robust evidence for the fundamental role of lipid metabolism during the GC reaction and unveiling a novel function for peroxisomes in B cells.Comments on revisions:There are still some discrepancies in gating strategies. In Fig. 7B legend (lines 1082-1083), they show representative flow plots of GL7+ CD95+ GC B cells among viable B cells, so it is not clear if they are IgDneg, as the rest of the GC B cells aforementioned in the text.

We apologize for missing this item in need of correction in the revision and sincerely thank the reviewer for the stamina and care in picking this up. The data shown in Fig. 7B represented cells (events) in the IgD^neg^ Dump^neg^ viable lymphoid gate. We will correct this omission/blemish in the final revision that becomes the version of record.

Western blot confirmation: We understand the limitations the authors enumerate. Perhaps an RT-qPCR analysis of the Dhrs7b gene in sorted GC B cells from the S1PR2-CreERT2 model could be feasible, as it requires a smaller number of cells. In any case, we agree with the authors that the results obtained using the huCD20-CreERT2 model are consistent with those from the S1PR2-CreERT2 model, which adds credibility to the findings and supports the conclusion that GC B cells in the S1PR2-CreERT2 model are indeed deficient in PexRAP.

We will make efforts to go back through the manuscript and highlight this limitation to readers, i.e., that we were unable to get genetic evidence to assess what degree of "counter-selection" applied to GC B cells in our experiments.

We agree with the referee that optimally to support the Imaging Mass Spectrometry (IMS) data showing perturbations of various ether lipids within GC after depletion of PexRAP, it would have been best if we could have had a qRT2-PCR that allowed quantitation of the Dhrs7b-encoded mRNA in flow-purified GC B cells, or the extent to which the genomic DNA of these cells was in deleted rather than 'floxed' configuration.

While the short half-life of ether lipid species leads us to infer that the enzymatic function remains reduced/absent, it definitely is unsatisfying that the money for experiments ran out in June and the lab members had to move to new jobs.

Lines 222-226: We believe the correct figure is 4B, whereas the text refers to 4C.

As for the 1st item, we apologize and will correct this error.

Supplementary Figure 1 (line 1147): The figure title suggests that the data on T-cell numbers are from mice in a steady state. However, the legend indicates that the mice were immunized, which means the data are not from steady-state conditions.

We will change the wording both on line 1147 and 1152.

**Reviewer #2 (Public review):**
Summary:In this study, Cho et al. investigate the role of ether lipid biosynthesis in B cell biology, particularly focusing on GC B cell, by inducible deletion of PexRAP, an enzyme responsible for the synthesis of ether lipids.Strengths:Overall, the data are well-presented, the paper is well-written and provides valuable mechanistic insights into the importance of PexRAP enzyme in GC B cell proliferation.Weaknesses:More detailed mechanisms of the impaired GC B cell proliferation by PexRAP deficiency remain to be further investigated. In minor part, there are issues for the interpretation of the data which might cause confusions by readers.Comments on revisions:The authors improved the manuscript appropriately according to my comments.

To re-summarize, we very much appreciate the diligence of the referees and Editors in re-reviewing this work at each cycle and helping via constructive peer review, along with their favorable comments and overall assessments. The final points will be addressed with minor edits since there no longer is any money for further work and the lab people have moved on.

The following is the authors’ response to the original reviews.

**Reviewer #1 (Public review):**
In this manuscript, Sung Hoon Cho et al. presents a novel investigation into the role of PexRAP, an intermediary in ether lipid biosynthesis, in B cell function, particularly during the Germinal Center (GC) reaction. The authors profile lipid composition in activated B cells both in vitro and in vivo, revealing the significance of PexRAP. Using a combination of animal models and imaging mass spectrometry, they demonstrate that PexRAP is specifically required in B cells. They further establish that its activity is critical upon antigen encounter, shaping B cell survival during the GC reaction.Mechanistically, they show that ether lipid synthesis is necessary to modulate reactive oxygen species (ROS) levels and prevent membrane peroxidation.Highlights of the Manuscript:The authors perform exhaustive imaging mass spectrometry (IMS) analyses of B cells, including GC B cells, to explore ether lipid metabolism during the humoral response. This approach is particularly noteworthy given the challenge of limited cell availability in GC reactions, which often hampers metabolomic studies. IMS proves to be a valuable tool in overcoming this limitation, allowing detailed exploration of GC metabolism.The data presented is highly relevant, especially in light of recent studies suggesting a pivotal role for lipid metabolism in GC B cells. While these studies primarily focus on mitochondrial function, this manuscript uniquely investigates peroxisomes, which are linked to mitochondria and contribute to fatty acid oxidation (FAO). By extending the study of lipid metabolism beyond mitochondria to include peroxisomes, the authors add a critical dimension to our understanding of B cell biology.Additionally, the metabolic plasticity of B cells poses challenges for studying metabolism, as genetic deletions from the beginning of B cell development often result in compensatory adaptations. To address this, the authors employ an acute loss-of-function approach using two conditional, cell-type-specific gene inactivation mouse models: one targeting B cells after the establishment of a pre-immune B cell population (Dhrs7b^f/f, huCD20-CreERT2) and the other during the GC reaction (Dhrs7b^f/f; S1pr2-CreERT2). This strategy is elegant and well-suited to studying the role of metabolism in B cell activation.Overall, this manuscript is a significant contribution to the field, providing robust evidence for the fundamental role of lipid metabolism during the GC reaction and unveiling a novel function for peroxisomes in B cells.

We appreciate these positive reactions and response, and agree with the overview and summary of the paper's approaches and strengths.

However, several major points need to be addressed:Major Comments:Figures 1 and 2The authors conclude, based on the results from these two figures, that PexRAP promotes the homeostatic maintenance and proliferation of B cells. In this section, the authors first use a tamoxifen-inducible full Dhrs7b knockout (KO) and afterwards Dhrs7bΔ/Δ-B model to specifically characterize the role of this molecule in B cells. They characterize the B and T cell compartments using flow cytometry (FACS) and examine the establishment of the GC reaction using FACS and immunofluorescence. They conclude that B cell numbers are reduced, and the GC reaction is defective upon stimulation, showing a reduction in the total percentage of GC cells, particularly in the light zone (LZ).The analysis of the steady-state B cell compartment should also be improved. This includes a more detailed characterization of MZ and B1 populations, given the role of lipid metabolism and lipid peroxidation in these subtypes.Suggestions for Improvement:B Cell compartment characterization: A deeper characterization of the B cell compartment in non-immunized mice is needed, including analysis of Marginal Zone (MZ) maturation and a more detailed examination of the B1 compartment. This is especially important given the role of specific lipid metabolism in these cell types. The phenotyping of the B cell compartment should also include an analysis of immunoglobulin levels on the membrane, considering the impact of lipids on membrane composition.

Although the manuscript is focused on post-ontogenic B cell regulation in Ab responses, we believe we will be able to polish a revised manuscript through addition of results of analyses suggested by this point in the review: measurement of surface IgM on and phenotyping of various B cell subsets, including MZB and B1 B cells, to extend the data in Supplemental Fig 1H and I. Depending on the level of support, new immunization experiments to score Tfh and analyze a few of their functional molecules as part of a B cell paper may be feasible.

Addendum / update of Sept 2025: We added new data with more on MZB and B1 B cells, surface IgM, and on Tfh populations.

GC Response Analysis Upon Immunization: The GC response characterization should include additional data on the T cell compartment, specifically the presence and function of Tfh cells. In Fig. 1H, the distribution of the LZ appears strikingly different. However, the authors have not addressed this in the text. A more thorough characterization of centroblasts and centrocytes using CXCR4 and CD86 markers is needed.The gating strategy used to characterize GC cells (GL7+CD95+ in IgD− cells) is suboptimal. A more robust analysis of GC cells should be performed in total B220+CD138− cells.

We first want to apologize the mislabeling of LZ and DZ in Fig 1H. The greenish-yellow colored region (GL7^+^ CD35^+^) indicate the DZ and the cyan-colored region (GL7^+^ CD35^+^) indicates the LZ. Addendum / update of Sept 2025: We corrected the mistake, and added new experimental data using the CD138 marker to exclude preplasmablasts.

As a technical note, we experienced high background noise with GL7 staining uniquely with PexRAP deficient (Dhrs7b^f/f^; Rosa26-CreER^T2^) mice (i.e., not WT control mice). The high background noise of GL7 staining was not observed in B cell specific KO of PexRAP (Dhrs7b^f/f^; huCD20-CreER^T2^). Two formal possibilities to account for this staining issue would be if either the expression of the GL7 epitope were repressed by PexRAP or the proper positioning of GL7^+^ cells in germinal center region were defective in PexRAPdeficient mice (e.g., due to an effect on positioning cues from cell types other than B cells). In a revised manuscript, we will fix the labeling error and further discuss the GL7 issue, while taking care not to be thought to conclude that there is a positioning problem or derepression of GL7 (an activation antigen on T cells as well as B cells).

While the gating strategy for an overall population of GC B cells is fairly standard even in the current literature, the question about using CD138 staining to exclude early plasmablasts (i.e., analyze B220^+^ CD138^neg^ vs B220^+^ CD138^+^) is interesting. In addition, some papers like to use GL7^+^ CD38^neg^ for GC B cells instead of GL7^+^ Fas (CD95)^+^, and we thank the reviewer for suggesting the analysis of centroblasts and centrocytes. For the revision, we will try to secure resources to revisit the immunizations and analyze them for these other facets of GC B cells (including CXCR4/CD86) and for their GL7^+^ CD38^neg^. B220^+^ CD138^-^ and B220^+^ CD138^+^ cell populations.

We agree that comparison of the Rosa26-CreERT2 results to those with B cell-specific lossof-function raise a tantalizing possibility that Tfh cells also are influenced by PexRAP. Although the manuscript is focused on post-ontogenic B cell regulation in Ab responses, we hope to add a new immunization experiments that scores Tfh and analyzes a few of their functional molecules could be added to this B cell paper, depending on the ability to wheedle enough support / fiscal resources.

Addendum / update of Sept 2025: Within the tight time until lab closure, and limited $$, we were able to do experiments that further reinforced the GC B cell data - including stains for DZ vs LZ sub-subsetting - and analyzed Tfh cells. We were not able to explore changes in functional antigenic markers on the GC B or Tfh cells.

The authors claim that Dhrs7b supports the homeostatic maintenance of quiescent B cells in vivo and promotes effective proliferation. This conclusion is primarily based on experiments where CTV-labeled PexRAP-deficient B cells were adoptively transferred into μMT mice (Fig. 2D-F). However, we recommend reviewing the flow plots of CTV in Fig. 2E, as they appear out of scale. More importantly, the low recovery of PexRAP-deficient B cells post-adoptive transfer weakens the robustness of the results and is insufficient to conclusively support the role of PexRAP in B cell proliferation in vivo.

In the revision, we will edit the text and try to adjust the digitized cytometry data to allow more dynamic range to the right side of the upper panels in Fig. 2E, and otherwise to improve the presentation of the in vivo CTV result. However, we feel impelled to push back respectfully on some of the concern raised here. First, it seems to gloss over the presentation of multiple facets of evidence. The conclusion about maintenance derives primarily from Fig. 2C, which shows a rapid, statistically significant decrease in B cell numbers (extending the finding of Fig. 1D, a more substantial decrease after a bit longer a period). As noted in the text, the rate of de novo B cell production does not suffice to explain the magnitude of the decrease.

In terms of proliferation, we will improve presentation of the Methods but the bottom line is that the recovery efficiency is not bad (comparing to prior published work) inasmuch as transferred B cells do not uniformly home to spleen. In a setting where BAFF is in ample supply in vivo, we transferred equal numbers of cells that were equally labeled with CTV and counted B cells. The CTV result might be affected by lower recovered B cell with PexRAP deficiency, generally, the frequencies of CTV^low^ divided population are not changed very much. However, it is precisely because of the pitfalls of in vivo analyses that we included complementary data with survival and proliferation in vitro. The proliferation was attenuated in PexRAP-deficient B cells in vitro; this evidence supports the conclusion that proliferation of PexRAP knockout B cells is reduced. It is likely that PexRAP deficient B cells also have defect in viability in vivo as we observed the reduced B cell number in PexRAP-deficient mice. As the reviewer noticed, the presence of a defect in cycling does, in the transfer experiments, limit the ability to interpret a lower yield of B cell population after adoptive transfer into µMT recipient mice as evidence pertaining to death rates. We will edit the text of the revision with these points in mind.

In vitro stimulation experiments: These experiments need improvement. The authors have used anti-CD40 and BAFF for B cell stimulation; however, it would be beneficial to also include antiIgM in the stimulation cocktail. In Fig. 2G, CTV plots do not show clear defects in proliferation, yet the authors quantify the percentage of cells with more than three divisions. These plots should clearly display the gating strategy. Additionally, details about histogram normalization and potential defects in cell numbers are missing. A more in-depth analysis of apoptosis is also required to determine whether the observed defects are due to impaired proliferation or reduced survival.

As suggested by reviewer, testing additional forms of B cell activation can help explore the generality (or lack thereof) of findings. We plan to test anti-IgM stimulation together with anti-CD40 + BAFF as well as anti-IgM + TLR7/8, and add the data to a revised and final manuscript.

Addendum / update of Sept 2025: The revision includes results of new experiments in which anti-IgM was included in the stimulation cocktail, as well as further data on apoptosis and distinguishing impaired cycling / divisions from reduced survival .

With regards to Fig. 2G (and 2H), in the revised manuscript we will refine the presentation (add a demonstration of the gating, and explicate histogram normalization of FlowJo).

It is an interesting issue in bioscience, but in our presentation 'representative data' really are pretty representative, so a senior author is reminded of a comment Tak Mak made about a reduction (of proliferation, if memory serves) to 0.7 x control. [His point in a comment to referees at a symposium related that to a salary reduction by 30% : A mathematical alternative is to point out that across four rounds of division for WT cells, a reduction to 0.7x efficiency at each cycle means about 1/4 as many progeny.]

We will try to edit the revision (Methods, Legends, Results, Discussion) to address better the points of the last two sentences of the comment, and improve the details that could assist in replication or comparisons (e.g., if someone develops a PexRAP inhibitor as potential therapeutic).

For the present, please note that the cell numbers at the end of the cultures are currently shown in Fig 2, panel I. Analogous culture results are shown in Fig 8, panels I, J, albeit with harvesting at day 5 instead of day 4. So, a difference of ≥ 3x needs to be explained. As noted above, a division efficiency reduced to 0.7x normal might account for such a decrease, but in practice the data of Fig. 2I show that the number of PexRAP-deficient B cells at day 4 is similar to the number plated before activation, and yet there has been a reasonable amount of divisions. So cell numbers in the culture of mutant B cells are constant because cycling is active but decreased and insufficient to allow increased numbers ("proliferation" in the true sense) as programmed death is increased. In line with this evidence, Fig 8G-H document higher death rates [i.e., frequencies of cleaved caspase3^+^ cell and Annexin V^+^ cells] of PexRAP-deficient B cells compared to controls. Thus, the in vitro data lead to the conclusion that both decreased division rates and increased death operate after this form of stimulation.

An inference is that this is the case in vivo as well - note that recoveries differed by ~3x (Fig. 2D), and the decrease in divisions (presentation of which will be improved) was meaningful but of lesser magnitude (Fig. 2E, F).

**Reviewer #2 (Public review):**
Summary:In this study, Cho et al. investigate the role of ether lipid biosynthesis in B cell biology, particularly focusing on GC B cell, by inducible deletion of PexRAP, an enzyme responsible for the synthesis of ether lipids.Strengths:Overall, the data are well-presented, the paper is well-written and provides valuable mechanistic insights into the importance of PexRAP enzyme in GC B cell proliferation.

We appreciate this positive response and agree with the overview and summary of the paper's approaches and strengths.

Weaknesses:More detailed mechanisms of the impaired GC B cell proliferation by PexRAP deficiency remain to be further investigated. In the minor part, there are issues with the interpretation of the data which might cause confusion for the readers.

Issues about contributions of cell cycling and divisions on the one hand, and susceptibility to death on the other, were discussed above, amplifying on the current manuscript text. The aggregate data support a model in which both processes are impacted for mature B cells in general, and mechanistically the evidence and work focus on the increased ROS and modes of death. Although the data in Fig. 7 do provide evidence that GC B cells themselves are affected, we agree that resource limitations had militated against developing further evidence about cycling specifically for GC B cells. We will hope to be able to obtain sufficient data from some specific analysis of proliferation in vivo (e.g., Ki67 or BrdU) as well as ROS and death ex vivo when harvesting new samples from mice immunized to analyze GC B cells for CXCR4/CD86, CD38, CD138 as indicated by Reviewer 1. As suggested by Reviewer 2, we will further discuss the possible mechanism(s) by which proliferation of PexRAP-deficient B cells is impaired. We also will edit the text of a revision where to enhance clarity of data interpretation - at a minimum, to be very clear that caution is warranted in assuming that GC B cells will exhibit the same mechanisms as cultures in vitro-stimulated B cells.

Addendum / update of Sept 2025: We were able to obtain results of intravital BrdU incorporation into GC B cells to measure cell cycling rates. The revised manuscript includes these results as well as other new data on apoptosis / survival, while deleting the data about CD138 populations whose interpretation was reasonably questioned by the referees.

**Reviewer #1 (Recommendations for the authors):**
We believe the evidence presented to support the role of PexRAP in protecting B cells from cell death and promoting B cell proliferation is not sufficiently robust and requires further validation in vivo. While the study demonstrates an increase in ether lipid content within the GC compartment, it also highlights a reduction in mature B cells in PexRAP-deficient mice under steady-state conditions. However, the IMS results (Fig. 3A) indicate that there are no significant differences in ether lipid content in the naïve B cell population. This discrepancy raises an intriguing point for discussion: why is PexRAP critical for B cell survival under steady-state conditions?

We thank the referee for all their care and input, and we agree that further intravital analyses could strengthen the work by providing more direct evidence of impairment of GC B cells in vivo. To revise and improve this manuscript before creation of a contribution of record, we performed new experiments to the limit of available funds and have both (i) added these new data and (ii) sharpened the presentation to correct what we believe to be one inaccurate point raised in the review.

(A) Specifically, we immunized mice with a B cell-specific depletion of PexRAP (Dhrs7b^D/D-B^ mice) and measured a variety of readouts of the GC B cells' physiology in vivo: proliferation by intravital incorporation of BrdU, ROS in the viable GC B cell gate, and their cell death by annexin V staining directly ex vivo. Consistent with the data with in vitro activated B cells, these analyses showed increased ROS (new - Fig. 7D) and higher frequencies of Annexin V^+^ 7AAD^+^ in GC B cells (GL7^+^ CD38^-^ B cell-gate) of immunized Dhrs7b^D/D-B^ mice compared with WT controls (huCD20-CreERT2^+/-^, Dhrs7b^+/+^) (new - Fig. 7E). Collectively, these results indicate that PexRAP aids (directly or indirectly) in controlling ROS in GC B cells and reduces B cell death, likely contributing to the substantially decreased overall GC B cell population. These new data are added to the revised manuscript in Figure 7.

Moreover, in each of two independent experiments (each comprising 3 vs 3 immunized mice), BrdU^+^ events among GL7^+^ CD38^-^ (GC B cell)-gated cells were reduced in the B cell-specific PexRAP knockouts compared with WT controls (new, Fig. 7F and Supplemental Fig 6E). This result on cell cycle rates in vivo is presented with caution in the revised manuscript text because the absolute labeling fractions were somewhat different in Expt 1 vs Expt 2. This situation affords a useful opportunity to comment on the culture of "P values" and statistical methods. It is intriguing to consider how many successful drugs are based on research published back when the standard was to interpret a result of this sort more definitively despite a merged "P value" that was not a full 2 SD different from the mean. In the optimistic spirit of the eLife model, it can be for the attentive reader to decide from the data (new, Fig. 7F and Supplemental Fig 6E) whether to interpret the BrdU results more strongly that what we state in the revised text.

(B) On the issue of whether or not the loss of PexRAP led to perturbations of the lipidome of B cells prior to activation, we have edited the manuscript to do a better job making this point more clear.

We point out to readers that in the resting, pre-activation state abnormalities were detected in naive B cells, not just in activated and GC B cells. In brief, the IMS analysis and LC-MS-MS analysis detected statistically significant differences in some, but not all, the ether phospholipids species in PexRAP deficient cells (some of which was in Supplemental Figure 2 of the original version).

With this appropriate and helpful concern having been raised, we realize that this important point merited inclusion in the main figures. We point specifically to a set of phosphatidyl choline ions shown in Fig. 3 (revised - panels A, B, D) of the revised manuscript (PC O-36:5; PC O-38:5; PC O-40:6 and -40:7).

For this ancillary record (because a discourse on the limitations of each analysis), we will note issues such as the presence of many non-B cells in each pixel of the IMS analyses (so that some or many "true positives" will fail to achieve a "significant difference") and for the naive B cells, differential rates of synthesis, turnover, and conversion (e.g., addition of another 2-carbon unit or saturation / desaturation of one side-chain). To the extent the concern reflects some surprise and perhaps skepticism that what seem relatively limited differences (many species appear unaffected, etc), we share in the sentiment. But the basic observation is that there are differences, and a reasonable connection between the altered lipid profile and evidence of effects on survival or proliferation (i.e., integration of survival and cell cycling / division).

Additionally, it would be valuable to evaluate the humoral response in a T-independent setting. This would clarify whether the role of PexRAP is restricted to GC B cells or extends to activated B cells in general.

We agree that this additional set of experiments would be nice and would extend work incrementally by testing the generality of the findings about Ab responses. The practical problem is that money and time ran out while testing important items that strengthen the evidence about GC B cells.

Finally, the manuscript would benefit from a thorough revision to improve its readability and clarity. Including more detailed descriptions of technical aspects, such as the specific stimuli and time points used in analyses, would greatly enhance the flow and comprehension of the study. Furthermore, the authors should review figure labeling to ensure consistency throughout the manuscript, and carefully cite the relevant references. For instance, S1PR2 CreERT2 mouse is established by Okada and Kurosaki (Shinnakasu et al ,Nat. Immunol, 2016)

We appreciate this feedback and comment, inasmuch as both the clarity and scholarship matter greatly to us for a final item of record. For the revision, we have given our best shot to editing the text in the hopes of improved clarity, reduction of discrepancies (helpfully noted in the Minor Comments), and further detail-rich descriptions of procedures. We also edited the figure labeling to give a better consistency. While we note that the appropriate citation of Shinnakasu et al (2016) was ref. #69 of the original and remains as a citation, we have rechecked other referencing and try to use citations with the best relevant references.

Minor Comments: The labeling of plots in Fig. 2 should be standardized. For example, in Fig. 2C, D, and G, the same mouse strain is used, yet the Cre+ mouse is labeled differently in each plot.

We agree and have tried to tighten up these features in the panels noted as well as more generally (e.g., Fig. 4, 5, 6, 7, 9; consistency of huCD20-CreERT2 / hCD20CreERT2).

According to the text, the results shown in Fig. 1G and H correspond to a full KO (Dhrs7b^f/f; Rosa26-CreERT2 mice). However, Fig. 1H indicates that the bottom image corresponds to Dhrs7b^f/f, huCD20-CreERT2 mice (Dhrs7bΔ/Δ -B).

We have corrected Fig. 1H to be labeled as Dhrs7b^Δ/Δ^ (with the data on Dhrs7b^Δ/Δ-B^ presented in Supplemental Figure 4A, which is correctly labeled). Thank you for picking up this error that crept in while using copy/paste in preparation of figure panels and failing to edit out the "-B"!

Similarly, the gating strategy for GC cells in the text mentions IgD− cells, while the figure legend refers to total viable B cells. These discrepancies need clarification.

We believe we located and have corrected this issue in the revised manuscript.

Figures 3 and 4. The authors claim that B cell expression of PexRAP is required to achieve normal concentrations of ether phospholipids.Suggestions for Improvement:Lipid Metabolism Analysis: The analysis in Fig. 3 is generally convincing but could be strengthened by including an additional stimulation condition such as anti-IgM plus antiCD40. In Fig. 4C, the authors display results from the full KO model. It would be helpful to include quantitative graphs summarizing the parameters displayed in the images.

We have performed new experiments (anti-IgM + anti-CD40) and added the data to the revised manuscript (new - Supplemental Fig. 2H and Supplemental Fig 6, D & F). Conclusions based on the effects are not changed from the original.

As a semantic comment and point of scientific process, any interpretation ("claim") can - by definition - only be taken to apply to the conditions of the experiment. Nonetheless, it is inescapable that at least for some ether P-lipids of naive, resting B cells, and for substantially more in B cells activated under the conditions that we outline, B cell expression of PexRAP is required.

With regards to the constructive suggestion about a new series of lipidomic analyses, we agree that for activated B cells it would be nice and increase insight into the spectrum of conditions under which the PexRAP-deficient B cells had altered content of ether phospholipids. However, in light of the costs of metabolomic analyses and the lack of funds to support further experiments, and the accuracy of the point as stated, we prioritized the experiments that could fit within the severely limited budget.

One can add that our results provide a premise for later work to analyze a time course after activation, and to perform isotopomer (SIRM) analyses with [13] C-labeled acetate or glucose, so as to understand activation-induced increases in the overall To revise the manuscript, we did however extrapolate from the point about adding BCR cross-linking to anti-CD40 as a variant form of activating the B cells for measurements of ROS, population growth, and rates of division (CTV partitioning). The results of these analyses, which align with and thereby strengthen the conclusions about these functional features from experiments with anti-CD40 but no anti-IgM, are added to Supplemental Fig 2H and Supplemental Fig 6D, F.

Figures 5, 6, and 7The authors claim that Dhrs7b in B cells shapes antibody affinity and quantity. They use two mouse models for this analysis: huCD20-CreERT2 and Dhrs7b f/f; S1pr2-CreERT2 mice.Suggestions for Improvement:Adaptive immune response characterization: A more comprehensive characterization of the adaptive immune response is needed, ideally using the Dhrs7b f/f; S1pr2-CreERT2 model. This should include: Analysis of the GC response in B220+CD138− cells. Class switch recombination analysis. A detailed characterization of centroblasts, centrocytes, and Tfh populations. Characterization of effector cells (plasma cells and memory cells).

Within the limits of time and money, we have performed new experiments prompted by this constructive set of suggestions.

Specifically, we analyzed the suggested read-outs in the huCD20-CreERT2, Dhrs7b^f/f^ model after immunization, recognizing that it trades greater signal-noise for the fact that effects are due to a mix of the impact on B cells during clonal expansion before GC recruitment and activities within the GC. In brief, the results showed that

(a) the GC B cell population - defined as CD138^neg^ GL7^+^ CD38^lo/neg^ IgD^neg^ B cells - was about half as large for PexRAP-deficient B cells net of any early- or preplasmablasts (CD138^+^ events) (new - Fig 5G);

(b) the frequencies of pre- / early plasmablasts (CD138^+^ GL7^+^ CD38^neg^) events (see new - Fig. 6H, I; also, new Supplemental Fig 5D) were so low as to make it unlikely that our data with the S1pr2-CreERT2 model (in Fig 7B, C) would be affected meaningfully by analysis of the CD138 levels;

(c) There was a modest decrease in centrocytes (LZ) but not centroblasts (DZ) (new - Fig 5H, I) - consistent with the immunohistochemical data of Supplemental Fig. 5A-C.

Because of time limitations (the "shelf life" of funds and the lab) and insufficient stock of the S1pr2-CreERT2, Dhrs7b^f/f^ mice as well as those that would be needed as adoptive transfer recipients because of S1PR2 expression in (GC-)Tfh, the experiments were performed instead with the huCD20-CreERT2, Dhrs7b^f/f^ model. We would also note that using this Cre transgene better harmonizes the centrocyte/centroblast and Tfh data with the existing data on these points in Supplemental Fig. 4.

(d) Of note, the analyses of Tfh and GC-Tfh phenotype cells using the huCD20-CreERT2 B cell type-specific inducible Cre system to inactivate Dhrs7b (new - Supplemental Fig 1G-I; which, along with new - Supplemental Fig 5E) provide evidence of an abnormality that must stem from a function or functions of PexRAP in B cells, most likely GC B cells. Specifically, it is known that the GC-Tfh population proliferates and is supported by the GC B cells, and the results of B cell-specific deletion show substantial reductions in Tfh cells both the GC-Tfh gating and the wider gate for plots of CXCR5/PD-1/ fluorescence of CD4 T cells

Timepoint Consistency: The NP response (Fig. 5) is analyzed four weeks postimmunization, whereas SRBC (Supp. Fig. 4) and Fig. 7 are analyzed one week or nine days post-immunization. The NP system analysis should be repeated at shorter timepoints to match the peak GC reaction.

This comment may stem from a misunderstanding. As diagrammed in Fig. 5A, the experiments involving the NP system were in fact measured at 7 d after a secondary (booster) immunization. That timing is approximately the peak period and harmonizes with the 7 d used for harvesting SRBC-immunized mice. So in fact the data with each system were obtained at a similar time point. Of course the NP experiments involved a second immunization so that many plasma cell and Ab responses derived from memory B cells generated by the primary immunization. However, the field at present is dominated by the view that the vast majority of the GC B cells after this second immunization (which historically we perform with alum adjuvant) are recruited from the naive rather than the memory B cell pool. For the revised manuscript, we have taken care that the Methods, Legend, and Figure provide the information to readers, and expanded the statement of a rationale.

It may seem a technicality but under NIH regulations we are legally obligated to try to minimize mouse usage. It also behooves researchers to use funds wisely. In line with those imperatives, we used systems that would simultaneously allow analyses of GC B cells, identification of affinity maturation (which is minimal in our hands at a 7 d time point after primary NP-carrier immunization), and a switched repertoire (also minimal), and where with each immunogen the GC were scored at 7-9 d after immunization (9 d refers to the S1pr2-CreERT2 experiments). Apart from the end of funding, we feel that what little might be learned from performing a series of experiments that involve harvests 7 d after a primary immunization with NP-ovalbumin cannot well be justified.

In vitro plasma cell differentiation: Quantification is missing for plasma cell differentiation in vitro (Supp. Fig. 4). The stimulus used should also be specified in the figure legend. Given the use of anti-CD40, differentiation towards IgG1 plasma cells could provide additional insights.

As suggested by reviewer, we have added the results of quantifying the in vitro plasma cell differentiation in Supplemental Fig 6B. Also, we edited the Methods and Supplemental Figure Legend to give detailed information of in vitro stimulation.

Proliferation and apoptosis analysis: The observed defects in the humoral response should be correlated with proliferation and apoptosis analyses, including Ki67 and Caspase markers.

As suggested by the review, we have performed new experiment and analyzed the frequencies of cell death by annexin V staining, and elected to use intravital uptake of BrdU as a more direct measurement of S phase / cell cycling component of net proliferation. The new results are now displayed in Figure 5 and Supplemental Fig. 5.

Western blot confirmation: While the authors have demonstrated the absence of PexRAP protein in the huCD20-CreERT2 model, this has not been shown in GC B cells from the Dhrs7b f/f; S1pr2-CreERT2 model. This confirmation is necessary to validate the efficiency of Dhrs7b deletion.

We were unable to do this for technical reasons expanded on below. For the revision, we have edited in a bit of text more explicitly to alert readers to the potential impact of counter-selection on interpretation of the findings with GC B cells. Before entering the GC, B cells have undergone many divisions, so if there were major pre-GC counterselection, in all likelihood the GC B cells would PexRAP-sufficient. To recap from the original manuscript and the new data we have added, IMS shows altered lipid profiles in the GC B cells and the literature indicates that the lipids are short-lived, requiring de novo resynthesis. The BrdU, ROS, and annexin V data show that GC B cells are abnormal. Accordingly, abnormal GC B cells represent the parsimonious or straightforward interpretation of the new results with GC-Tfh cell prevalence.

While we take these findings together to suggest that counterselection (i.e., a Western result showing normal levels of PexRAP in the GC B cells) seems unlikely, it is formally possible and would mean that the in situ defects of GC B cells arose due to environmental influences of the PexRAP-deficient B cells during the developmental history of the WT B cells observed in the GC.

Having noted all that, we understand that concerns about counter-selection are an issue if a reader accepts the data showing that mutant (PexRAP-deficient) B cells tend to proliferate less and die more readily. Indeed, one can speculate that were we also to perform competition experiments in which the Ighb, Cd45.2 B cells (WT or Dhrs7b D/D) are mixed with equal numbers of Igha, Cd45.1 competitors, the differences would become much greater. With this in mind, Western blotting of flow-purified GC B cells might give a sense of how much counter-selection has occurred.

That said, the Westerns need at least 2.5 x 10^6^ B cells (those in the manuscript used five million, 5 x 10^6^) and would need replication. Taken together with the observation that ~200,000 GC B cells (on average) were measured in each B cell-specific knockout mouse after immunization (Fig. 1, Fig 5) and taking into account yields from sorting, each Western would require some 20-25 tamoxifen-injected ___-CreERT2, Dhrs7b f/f mice, and about half again that number as controls. The expiry of funds prohibited the time and costs of generating that many mice (>70) and flow-purified GC B cells.

Figure 8The authors claim that Dhrs7b contributes to the modulation of ROS, impacting B cell proliferation.Suggestions for Improvement:GC ROS Analysis: The in vitro ROS analysis should be complemented by characterizing ROS and lipid peroxidation in the GC response using the Dhrs7b f/f; S1pr2-CreERT2 model. Flow cytometry staining with H2DCFDA, MitoSOX, Caspase-3, and Annexin V would allow assessment of ROS levels and cell death in GC B cells.

While subject to some of the same practical limits noted above, we have performed new experiments in line with this helpful input of the reviewer, and added the helpful new data to the revised manuscript. Specifically, in addition to the BrdU and phenotyping analyses after immunization of huCD20-CreER^T2^, Dhrs7b^f/f^ mice, DCFDA (ROS), MitoSox, and annexin V signals were measured for GC B cells. Although the mitoSox signals did not significantly differ for PexRAP-deficient GCB, the ROS and annexin V signals were substantially increased. We added the new data to Figure 5 and Supplemental Figure 5. Together with the decreased in vivo BrdU incorporation in GC B cells from Dhrs7b^D/D-B^ mice, these results are consistent with and support our hypothesis that PexRAP regulates B cell population growth and GC physiology in part by regulating ROS detoxification, survival and proliferation of B cells.

Quantification is missing in Fig. 8E, and Fig. 8F should use clearer symbols for better readability.

We added quantification for Fig 8E in Supplemental Fig 6E, and edited the symbols in Fig 8F for better readability.

Figure 9The authors claim that Dhrs7b in B cells affects oxidative metabolism and ER mass. The results in this section are well-performed and convincing.Suggestion for Improvement:Based on the results, the discussion should elaborate on the potential role of lipids in antigen presentation, considering their impact on mitochondria and ER function.

We very much appreciate the praise of the tantalizing findings about oxidative metabolism and ER mass, and will accept the encouragement that we add (prudently) to the Discussion section to make note of the points mentioned by the Reviewer, particularly now that (with their encouragement) we have the evidence that B cell-specific loss of PexRAP (with the huCD20-CreERT2 deletion prior to immunization) resulted in decreased (GC-)Tfh and somewhat lower GC B cell proliferation.

**Reviewer #2 (Recommendations for the authors):**
The authors should investigate whether PexRAP-deficient GC B cells exhibit increased mitochondrial ROS and cell death ex vivo, as observed in in vitro cultured B cells.

We very much appreciate the work of the referee and their input. We addressed this helpful recommendation, in essence aligned with points from Reviewer 1, via new experiments (until the money ran out) and addition of data to the manuscript. To recap briefly, we found increased ROS in GC B cells along with higher fractions of annexin V positive cells; intriguingly, increased mtROS (MitoSox signal) was not detected, which contrasts with the results in activated B cells in vitro in a small way. To keep the text focused and not stray too far outside the foundation supported by data, this point may align with papers that provide evidence of differences between pre-GC and GC B cells (for instance with lack of Tfam or LDHA in B cells).

It remains unclear whether the impaired proliferation of PexRAP-deficient B cells is primarily due to increased cell death. Although NAC treatment partially rescued the phenotype of reduced PexRAP-deficient B cell number, it did not restore them to control levels. Analysis of the proliferation capacity of PexRAP-deficient B cells following NAC treatment could provide more insight into the cause of impaired proliferation.

To add to the data permitting an assessment of this issue, we performed new experiments in which B cells were activated (BCR and CD40 cross-linking), cultured, and both the change in population and the CTV partitioning were measured in the presence or absence of NAC. The results, added to the revision as Supplemental Fig 6FH, show that although NAC improved cell numbers for PexRAP-deficient cells relative to controls, this compound did not increase divisions at all. We infer that the more powerful effect of this lipid synthesis enzyme is to promote survival rather than division capacity.

Primary antibody responses were assessed at only one time point (day 20). It would be valuable to examine the kinetics of antibody response at multiple time points (0, 1w, 2w, 3w, for example) to better understand the temporal impact of PexRAP on antibody production.

We thank the reviewer for this suggestion. While it may be that the kinetic measurement of Ag-specific antibody level across multiple time points would provide an additional mechanistic clue into the of impact PexRAP on antibody production, the end of sponsored funding and imminent lab closure precluded performing such experiments.

CD138+ cell population includes both GC-experienced and GC-independent plasma cells (Fig. 7). Enumeration of plasmablasts, which likely consists of both PexRAP-deleted and undeleted cells (Fig. 7D and E), may mislead the readers such that PexRAP is dispensable for plasmablast generation. I would suggest removing these data and instead examining the number of plasmablasts in the experimental setting of Fig. 4A (huCD20-CreERT2-mediated deletion) to address whether PexRAP-deficiency affects plasmablast generation.

We have eliminated the figure panels in question, since it is accurate that in the absence of a time-stamping or marking approach we have a limited ability to distinguish plasma cells that arose prior to inactivation of the Dhrs7b gene in B cells. In addition, we performed new experiments that were used to analyze the "early plasmablast" phenotype and added those data to the revision (Supplemental Fig 5D).